



# Estimated regional $CO_2$ flux and uncertainty based on an ensemble of atmospheric $CO_2$ inversions

Naveen Chandra[1], Prabir K. Patra[1,2,3], Yousuke Niwa[4], Akihiko Ito[4], Yosuke Iida[5], Daisuke Goto[6], Shinji Morimoto[3], Masayuki Kondo[7], Masayuki Takigawa[1], Tomohiro Hajima[1], Michio Watanabe[1]

[1]Research Institute for Global Change, JAMSTEC, 3173-25 Showa-machi, Kanazawa, Yokohama, 236-0001, Japan

[2]Center for Environmental Remote Sensing, Chiba University, Chiba, 263-8522, Japan

[3]Graduate School of Science, Tohoku University, 6-3 Aoba, Aramaki, Aoba-ku, Sendai 980-8578, Japan

[4]Earth System Division, National Institute for Environmental Studies, Tsukuba 305-8506, Japan

[5]Atmosphere and Ocean Department, Japan Meteorological Agency, Tokyo 105-8431, Japan

[6]National Institute of Polar Research, 10-3 Midori-cho, Tachikawa, Tokyo, 190-8518, Japan

[7]Institute for Space-Earth Environmental Research, Nagoya University, Nagoya, Aichi 464-8601 Japan

*Correspondence to*: Naveen Chandra (naveennegi@jamstec.go.jp) and Prabir K. Patra (prabir@jamstec.go.jp)

**Abstract.** Global and regional sources and sinks of carbon across the earth's surface have been studied extensively using atmospheric carbon dioxide ($CO_2$) observations and chemistry-transport model (ACTM) simulations (top-down/inversion method). However, the uncertainties in the regional flux (+ve: source to the atmosphere; -ve: sink on land/ocean) distributions remain unconstrained mainly due to the lack of sufficient high-quality measurements covering the globe in all seasons and the uncertainties in model simulations. Here, we use a suite of 16 inversion cases, derived from a single transport model (MIROC4-

ACTM) but different sets of a priori (bottom-up) terrestrial biosphere and oceanic fluxes, as well as prior flux and observational data uncertainties (50 sites) to estimate $CO_2$ fluxes for 84 regions over the period 2000-2020. The ensemble inversions provide a mean flux field that is consistent with the global $CO_2$ growth rate, land and ocean sink partitioning of -2.9±0.3 (±1σ uncertainty on mean) and -1.6±0.2 PgC yr$^{-1}$, respectively, for the period 2011-2020 (without riverine export correction), offsetting about 22-33% and 16-18% of global fossil-fuel $CO_2$ emissions. Aggregated fluxes for 15 land regions compare

reasonably well with the best estimations for (approx. 2000-2009) given by the REgional Carbon Cycle Assessment and Processes (RECCAP), and all regions appeared as a carbon sink over 2011-2020. Interannual variability and seasonal cycle in $CO_2$ fluxes are more consistently derived for different prior fluxes when a greater degree of freedom is given to the inversion system (greater prior flux uncertainty). We have evaluated the inversion fluxes using independent aircraft and surface measurements not used in the inversions, which raises our confidence in the ensemble mean flux rather than an individual

inversion. Differences between 5-year mean fluxes show promises and capability to track flux changes under ongoing and future $CO_2$ emission mitigation policies.



## 1. Introduction

Carbon dioxide is the most important anthropogenic greenhouse gas in the Earth's atmosphere. Due to human influences, e.g., fossil fuels consumption and cement production (FFC), the concentration of atmospheric $CO_2$ has

increased (by 38%) from 289.9±3.3 ppm in 1850-1900 to 398.8±7.3 ppm in 2010-2019, with the fastest growth in past five decades (Canadell et al., 2021). To limit global warming below 1.5°C by 2100, as per the Paris Agreement, a drastic and sustained reduction in $CO_2$ emissions from anthropogenic activities is recommended in the IPCC's 6[th] assessment report (AR6). The IPCC AR6 Working Group 1 estimated remaining carbon budgets (starting from 1 January 2020) for limiting global warming to 1.5°C, 1.7°C, and 2.0°C as 140 PgC, 230 PgC, and 370 PgC,

respectively, based on the 50[th] percentile of the transient climate response to cumulative carbon emissions (TCRE). With the present FFC emissions of about 10 PgC yr$^{-1}$ (Jones et al., 2021), the remaining carbon budget will be consumed within decades.

The sinks on the land and ocean constitute a major component of nature-based solutions (Canadell et al., 2021).

One of the impediments to making policy for $CO_2$ emission reduction is poor knowledge of the regional sources and sinks of carbon in Earth's disturbed and undisturbed ecosystems on land and in the ocean. During 2010–2019, the $CO_2$ emissions from human activities (average rate = 10.9±0.9 PgC yr$^{-1}$) was distributed between three Earth system components: 46% accumulated in the atmosphere (5.1±0.02 PgC yr$^{-1}$), 23% was taken up by the ocean (2.5±0.6 PgC yr$^{-1}$) and 31% was stored by vegetation in terrestrial ecosystems (3.4±0.9 PgC yr$^{-1}$) (Table 5.1 in

Canadell et al., 2021). Large uncertainties persist for the total global land and ocean sink partitioning in the IPCC assessment, up to about 25% of the global total land and ocean sinks. The uncertainty in land and ocean sink partitioning exists greatly in the Global Carbon Project (GCP)'s annual carbon budget release (Friedlingstein et al., 2020).

Top-down inverse models estimate residual natural or non-FFC $CO_2$ fluxes from land and ocean regions because inversion calculations do not optimise the FFC emissions. When different sources of FFC emissions are used in inversions, post-inversion corrections are applied for comparison between inversions (Peylin et al., 2013; Thompson et al., 2016). More recently, the inverse model inter-comparison experiments use prescribed FFC emissions, e.g., the global carbon project (Friedlingstein et al., 2020) or the OCO-2 flux intercomparison (Crowell

et al., 2019) to avoid the post-inversion correction. However, the impacts of biases in FFC emissions on inversion estimated $CO_2$ fluxes remained relatively unexplored (Saeki and Patra, 2017). The FFC emission biases affect the region of FFC emission and the regions linked closely by atmospheric transport. This is because 1) the prior flux


uncertainties set for each of the inversion regions may not be sufficient to allow fully compensatory correction by inversion, 2) the model transport biases could move FFC emission signals slower or faster from the source region, 3) emission signal goes undetected within the region when there are not enough measurement sites in the source region of biased FFC. Most FFC emission inventories are based on the data available on the International Energy Agency (IEA) and British Petroleum and mainly differ in spatial distribution within a given country (Crippa et al., 2020; Jones et al., 2021; Oda et al., 2018).

The GCP annual updates of inversions provide a metric for evaluating inversions using independent measurements, mainly from the aircraft campaigns (e.g., Friedlingstein et al., 2020). We argue in this work that the metric for evaluation of regional fluxes should be evaluated using a new transport model simulation of the predicted fluxes, not using the assimilated $CO_2$ field. This hypothesis is put forward because estimation of predicted fluxes from model-data differences may not be straightforward due to the underlying assumptions of the data assimilation systems for flux correlation lengths, observational data uncertainty, etc. (Baker et al., 2010; Chevallier et al., 2007; van der Laan-Luijkx et al., 2017; Miyazaki et al., 2011; Niwa et al., 2017; Rodenbeck et al., 2003). In other words, good statistics for the validation metric using independent data did not ensure good agreement between the estimated fluxes by different models, at the sub-hemispheric and sub-continental scales, or separately for land and ocean (Gaubert et al., 2019; Long et al., 2021). Another way of improving our knowledge about regional flux estimation uncertainties in regional fluxes is to employ multiple types of datasets from both bottom-up and top-down modelling systems (Kondo et al., 2020).

The uncertainties in the regional fluxes mainly arise from prior flux distribution and seasonality, selection of observational network and data uncertainty, transport model resolution leading site-representation error, uncertainties arising from parameterisation of transport processes. The uncertainties associated with the subcontinental scale $CO_2$ fluxes are often much greater than the interannual and interdecadal flux changes in non-FFC sectors, which allows us a better assessment of the changes in regional $CO_2$ fluxes compared to knowledge gained in regional flux magnitudes (Baker et al., 2006; Gurney et al., 2002; Patra et al., 2005a, 2005b; Peylin et al., 2013; Rödenbeck et al., 2003). Typically, multi-model assessment of flux estimation uncertainties is performed by collecting inversions from different transport models, e.g., in TransCom (Gurney et al., 2002; Peylin et al., 2013), for inversions using GOSAT measurements (Houweling et al., 2015) or for inversions using OCO-2 (Crowell et al., 2019). Such intercomparisons used single inversion from different modelling groups and provided the range in $CO_2$ flux uncertainty due to differences in transport models. However, no quantification of the flux uncertainties





due to the choices of prior fluxes distribution, prior flux uncertainty, and observational data uncertainty could be

obtained.

Here we show an ensemble-based inversion approach based on different choices of prior flux uncertainties and representation of measurement data uncertainties, using a single chemistry-transport model (JAMSTEC's MIROC4-ACTM). The details of the MIROC4-ACTM, observed and model data processing, inversion setup are

in section 2, followed by the Results and discussion. The fluxes and uncertainties are presented at regional and global scales, along with their validation using inversion-independent observations. Although the inversions are performed for the period 1998-2020, results are discussed mainly for the two most recent decades (2001-2010 and 2011-2020), with the only exception for comparing land-ocean flux partitioning with the IPCC AR6. Conclusions are given in section 4.


## 2. Materials and method

### 2.1. JAMSTEC's MIROC version 4 Atmospheric Chemistry-Transport Model (MIROC4-ACTM)

The Model for Interdisciplinary Research on Climate, version 4 (MIROC4; Watanabe et al., 2008) atmospheric general circulation model (AGCM)-based chemistry transport model (referred to as MIROC4-ACTM; Patra et al.,

2018) is used for the forward simulations of $CO_2$. The MIROC earth system model is developed at JAMSTEC in collaboration with the University of Tokyo and the National Institute of Environmental Studies (NIES) (Kawamiya et al., 2020). Simulations of long-lived gases in the atmosphere ($CO_2$, $CH_4$, $N_2O$, $SF_6$: (Bisht et al., 2021; Chandra et al., 2021; Patra et al., 2017, 2018) are performed at a horizontal resolution of T42 spectral truncations (~2.8° ×2.8° latitude-longitude grid) with 67 vertical hybrid-pressure layers between the Earth's surface and 0.0128 hPa

(~80 km). The simulated horizontal winds (U, V) and temperature (T) are nudged with the Japan Meteorological Agency Reanalysis data product (JRA-55; Kobayashi et al., 2015) at the altitude range of ~980-0.018 hPa for better representation of the atmospheric transport at synoptic and seasonal timescales. An accurate representation of transport is essential for performing inverse model calculation by minimising biases in horizontal and vertical gradients in the simulated tracer fields. We tested the large-scale interhemispheric transport and Brewer-Dobson

circulation in the MIROC4-ACTM using the $SF_6$ simulations in the troposphere and $CO_2$-derived age of air in the troposphere and stratosphere (Bisht et al., 2021; Patra et al., 2018). A close match between observed and modelled $SF_6$ and photochemically inert $CO_2$ vertical gradient in the troposphere and lower stratosphere manifests the





accurate transport in the MIROC4-ACTM. That enables us to use any mismatch between observation and simulations to estimate the land and oceanic fluxes using the inverse modelling technique (details in Section 2.4).


## 2.2 A priori $CO_2$ simulations

We simulated $CO_2$ tracers corresponding to the FFC ($CO_2^{ff}$), land biosphere fluxes ($CO_2^{lnd}$), fire emissions ($CO_2^{fire}$), and ocean exchanges ($CO_2^{ocn}$) from different sets of prior (bottom-up) emissions (Table 1). $CO_2^{ff}$ is simulated using the gridded fossil fuel emission dataset (GridFED; Jones et al., 2021). $CO_2^{lnd}$ tracers are simulated using two sets

of terrestrial biosphere fluxes from the Carnegie-Ames-Stanford Approach (CASA) biogeochemical model (Randerson et al., 1997)), and Vegetation Integrative Simulator for Trace Gases (VISIT) (Ito, 2019). The CASA fluxes are annually balanced, seasonally varying flux due to terrestrial photosynthesis and respiration, while the VISIT simulation accounts for $CO_2$ fertilisation (increasing photosynthesis due to rising atmospheric $CO_2$), LUC (perturbation on terrestrial carbon budget via land-use change), and climate variability (on the net a large land sink;

Table 1). The CASA and VISIT monthly-mean fluxes are downscaled to 3-hourly time intervals using JRA-55 meteorology (Table 1). Monthly-mean fire emissions are used from GFEDv4s (van der Werf et al., 2017) for simulating the $CO_2^{fire}$ tracer. Sea-air $CO_2$ fluxes are taken from an upscaling model of shipboard measurements of $pCO_2$ (referred to here as TT09: Takahashi et al., 2009), and an empirical model of the Japan Meteorological Agency (JMA; Iida et al., 2021) for simulating $CO_2^{ocn}$ tracers. The seasonal cycle for TT09 sea-air exchange flux

is stationary, like that of the CASA, over the analysis period, while the JMA oceanic $CO_2$ fluxes vary interannually, as in the case of VISIT. The model and prior fluxes' details are given in Table 1.

We prepare two cases of prior $CO_2$ simulations by adding the $CO_2$ flux tracers in different combinations as follows:

gc3t = $CO_2^{lnd}$ (CASA) + $CO_2^{ocn}$ (TT09) + $CO_2^{ff}$

gvjf = $CO_2^{lnd}$ (VISIT) + $CO_2^{ocn}$ (JMA) + $CO_2^{ff}$ + $CO_2^{fire}$

The gvjf case includes all the known interannual variability in land due to climate as simulated by VISIT and ocean fluxes by JMA. In contrast, the gc3t case has no information on interannual variability in land and ocean fluxes and the annual land sink. These two a priori flux cases are designed to evaluate the strength of MIROC4-ACTM

simulations to derive fluxes consistently (or the lack of it) given the information on $CO_2$ measurements from a network of sites and the statistics of prior flux uncertainty and measurement data uncertainty.



### 2.3 Atmospheric Data Selection and curve fitting

We used $CO_2$ observations from 50 measurement sites (marked in Fig. 1) for the inverse modelling (Supplementary
Table S1). The 38 sites are chosen from GML/NOAA, 4 from CSIRO, one from LSCE/IPSL, 2 from SIO-CO$_2$, 1
from SAWS, 1 from ECCC, 3 from JMA. Data until 2019 are taken from
obspack_co2_1_GLOBALVIEWplus_v6.1_2021-03-01 (Schuldt et al., 2021), and JMA data are taken from
WDCGG. Extension to obspack_co2_1_GLOBALVIEWplus_v6.1 for 2020 is compiled from GML/NOAA data
and WDCGG websites as appropriate. We further extended the 2020 values into 2021 based on the growth rate
determined for Jan-Mar observed at Minamitorishima (MNM) as available on the WDCGG (the results of 2021
will not be used in any analysis and will be treated as the spin-down year of inversion). The model simulations are
sampled at observation time and the nearest grid of observation location at hourly intervals. We selected the sites
where the temporal data gaps are minimum; no more than six months data gaps at a stretch for the inversion period
(1999–2020). These temporal data gaps (1-6 months) are filled using the curve fitting method based on the digital
filtering technique (Nakazawa et al., 1997). We fit the measured and simulated time-series at daily-weekly time
intervals with six harmonics by a cut-off length of 24 months for the digital filter.

### 2.4. Inverse method

Inverse analysis of atmospheric $CO_2$ helps to link the atmospheric observations to carbon fluxes from land and
ocean. We use a time-dependent Bayesian inversion system, initially developed by (Rayner et al., 1999). The
inversion formalism specifies prior estimates of both the fluxes and their uncertainty (called prior flux uncertainty
(PFU)) and optimises fluxes over 54 lands and 30 ocean regions (Fig. 1) by minimising the difference between the
$CO_2$ mixing ratios simulated by the MIROC4-ACTM and observed at 50 measurement sites for 1998-2021. We
exclude the first two years and last year of inversion from our analysis period (2000-2020) to avoid the edge effect.

In the Bayesian inversion, the cost function

$$(D - GS)^T C_D^{-1} (D - GS) + (S - S_o)^T C_{So}^{-1} (S - S_o) \qquad (1)$$

is minimised with the solution

$$S = S_o + (G^T C_D^{-1} G + C_{So}^{-1})^{-1} G^T C_D^{-1} (D_{obs} - D_{ACTM}) \qquad (2)$$

and posterior error covariance



$$C_s = (G^T C_D^{-1} G + C_{So}^{-1})^{-1} \qquad\qquad\qquad (3)$$

where $S_o$ is the regional prior source matrix, $C_{So}$ is a prior source covariance matrix, $D_{obs}$ is the measurement data matrix, $D_{ACTM}$ is forward model simulations using a priori fluxes and, $C_D$ is the data covariance matrix. The linear relationship between the regional emission (S) matrix and model-observation difference matrix (D) is defined by Green's function (G) using the unitary regional source to concentration change relationship for the 84 regions depicted in Fig. 1a. The elements in monthly flux (S) are the optimised fluxes (referred to as a posteriori or predicted flux) of $CO_2$ from 84 regions of the inverse model, and the elements of $C_D$ and $C_{So}$ are the error variance-covariance matrix of the observational data (D) and a priori source ($S_0$). The off-diagonal elements of $C_{So}$ are kept zero, assuming the a priori fluxes are uncorrelated to one another. The correction flux (second term in Eq. 2) is primarily determined by the difference between $D_{obs}$ and $D_{ACTM}$, scaled by the data uncertainty.

The inversion settings based on the choice of a priori fluxes, measurement data uncertainty (MDU), and prior flux uncertainty (PFU) are crucial for flux estimation. The MDU refers to the degree to which the predicted concentrations are required to be fitted by the inverse model. In addition to measurement precision, MDU incorporates the inability of coarse spatial-resolution global ACTMs to simulate the concentrations at the observation sites. PFU decides the degree of freedom or allowed flux adjustment for each of the 84 regions to match the atmospheric data. It determines to what extent the priors are relied upon to constrain the posterior flux estimates. To determine the robustness of our results, we have performed sensitivity analysis by varying PFU and MDU (Table 2). In the first approach, we assign uniformly 2 PgC yr$^{-1}$ uncertainty to each of the 54 land regions and 1 PgC yr$^{-1}$ to each of the 30 ocean regions (referred to as PFU = "ctl"). In the second approach, we assign the land uncertainty by scaling the regional total FLUXCOM gross primary productivity (GPP) (Jung et al., 2017), while the uncertainty for the ocean regions is kept at 0.5 PgC yr$^{-1}$. In this case, regional total GPPs were multiplied by 2, and the upper limit is set at 2 PgC yr$^{-1}$ (referred to as gpp_v2; PFUs varied from 0.2-2.0 PgC yr$^{-1}$). We construct two additional PFU cases (gpp_v3 and gpp_v4) by multiplying the regional total GPPs by a factor of 3 and 4, respectively, and the maximum allowed set at 3 and 4 PgC yr$^{-1}$. The land PFUs varied as 0.4-3.0 PgC yr$^{-1}$ and 0.6-4.0 PgC yr$^{-1}$ for the gpp_v3 and gpp_v4, respectively. The ocean PFUs were set at 0.75 PgC yr$^{-1}$ for gpp_v3 and 1.0 PgC yr$^{-1}$ for gpp_v4 (Table 2).

The monthly-mean residual standard deviation (RSD), from the difference between measured and fitted data, plus a constant value to account for the measurement accuracy are used for monthly varying measurement data uncertainty (MDU) at each station for inverse modelling calculations. The absolute magnitude of MDU is chosen



in such a way that the estimated flux is optimized to the data only to an appropriate level commensurate with the ability of ACTM to model them. We prepared two MDU cases by multiplying the RSDs by a factor of 2 (referred

to as "ux2") and 4 (referred to as "ux4"), and added to an estimated measurement uncertainty of 0.1 ppm. Based on different combinations of four PFU (ctl, gpp_v2, gpp_v3, gpp_v4), two MDU (ux2, ux4), and two prior flux cases (gc3t and gvjf), we run 16 set of inversion cases (Table 2). Based on the inversions with multiple priors, PFUs and MDUs, we will present estimated mean/median fluxes and spread as 1-$\sigma$ standard deviations from 16 ensemble members.

Figure 2 shows the examples of simulated and observed time series of $CO_2$ (top row), and simulated-observed differences (bottom row) at two selected sites, Alert and Syowa. The results show faster (slower) $CO_2$ increase rates for the a priori flux simulated case gc3t (gvjf), mainly because of no land sink in CASA flux and stronger land sink in VISIT flux (broken red and purple lines, respectively). By using the mean of these two a priori flux scenarios, the prior $CO_2$ concentrations show a better match with the $CO_2$ growth rates (refer to the grey lines in

the lower row; Fig. 2). Even after inversion, mild overestimation (underestimation) of the $CO_2$ growth rate for gc3t (gvjf) cases persisted, and by taking ensemble mean inversion flux the $CO_2$ growth rates are perfectly matched with the observations (Fig. 2), which is sometimes set as an evaluation metric for atmospheric $CO_2$ inversions (Friedlingstein et al., 2020).

**2.5 Performance of inversion**

We use flux uncertainty reduction (FUR) to identify which regions are well constrained by the data. FUR is a standard diagnostic of Bayesian estimation and defined by

$$FUR = 1 - \frac{\sigma_{predicted}}{\sigma_{prior}}$$

where $\sigma_{prior}$ and $\sigma_{predicted}$ represent the prior and predicted flux uncertainties. High values (FUR towards 100) indicate

strong data constraints, while low values (close to 0) indicate that the data are not able to move the estimates away from the prior. To identify which parts of the land and ocean have been constrained significantly by the inversions, PFU, predicted flux uncertainty, and FUR is plotted in Fig. 3 for the four PFU cases of gc3t and ux4 setup. The PFU case "gpp_v4" and "ctl" shows observational constraint over most of the region (grey shaded areas on the right column). Good constraints (larger FUR) are found in the Northern America, Russia, Southern Ocean, Tropical

and South Pacific, South Indian, North Atlantic, highlighting the large long-running observational programs by USA, Japanese, and European research groups. The poor constraints (low FUR) are observed over South Asia,



West Asia, Northern Africa. The Tropical Indian Ocean due to the lack of observations. It is also noted that FUR is influenced by PFU settings. For example, a smaller a priori uncertainty, i.e., gpp_v2 achieved lower FUR. As discussed later in this article, the FUR is only indicative of the observational constraint regional fluxes, and we

recommend that the spread of ensemble inversions provide more representative estimation of the regional $CO_2$ sources and sinks.

## 2.6 Flux processing and regional uncertainty estimations

The predicted fluxes from 84 regions (54 land and 30 oceans) are regridded into 1º x 1º spatial resolutions based

on the land and ocean basis functions. Once regridded, the fluxes were aggregated into 15 land and 11 ocean regions for further analysis (ref. Fig. 1b). We first estimated the mean of each inversion for different periods (monthly, annual, five-year, decadal) as per analysis. Then, we averaged the individual means (n=16) to estimate the ensemble mean and standard deviations. The ensemble mean (here and, in the following, referred to as "ensm") is the best estimate (i.e., a measure of central tendency) of land-air and sea-air exchange carbon flux. There are different

options to characterize "uncertainty" in $CO_2$ flux estimates, for example, the standard deviation, standard error, 95% confidence intervals, interquartile ranges. Here, we followed the standard deviation of the multi-inversion means as a metric of the "uncertainty" (i.e., variability) in the multi-inversion estimates.

## 2.7. Observations used for predicted fluxes validation

The predicted fluxes are validated by comparing the posterior 3-dimensional $CO_2$ observations to independent (i.e., not used in the inversions) aircraft observations. The aircraft observations used for validation include data published in obspack_co2_1_GLOBALVIEWplus_v6.1.

### 2.7.1 HIPPO and ATom observations

We used the $CO_2$ from two sets of aircraft campaigns: the HIAPER Pole-to-Pole Observations (HIPPO) during the period January 7, 2009, to September 15, 2011 (Wofsy, 2011) and the Atmospheric Tomography Missions (ATom) during the period July 29, 2016, to May 21, 2018 (Wofsy et al., 2018) to validate the latitude-altitude gradients covering different seasons over the Pacific and Atlantic Oceans. The four HIPPO campaigns (HIPPO-1 in January 2009, HIPPO-2 in October/December 2009, HIPPO-3 in March/April 2010 and HIPPO-4 in May/July 2011),

performed from 82°N to 67°S over the Pacific (but also partly cover the North American continent) and with continuous profiling between ~150 m and 8500 m altitudes at approximately 2.2º intervals, are used for the





validation. The ATom mission is built upon HIPPO mission, but with a wider horizontal extent with global coverage over the Pacific, the Atlantic, and the Arctic oceans. The comparisons performed for all the four ATom circuits occurred in July–August 2016 (ATom-1), January–February 2017 (ATom-2), September– October 2017

(ATom-3), and April–May 2018 (ATom-4). The mission consisted of 48 science flights and 548 vertical profiles over the Pacific and Atlantic basins.

### 2.7.2. NOAA measurements

We use NOAA regular aircraft-based vertical profiles also to validate the simulated vertical gradients in the

troposphere (Sweeney et al., 2015). The aircraft sites are located mainly over the North American continent. The aircraft profiles include measurements over the Southern Great Plains (SGP: 2006-2019), operated by the U.S. Department of Energy (Biraud et al., 2013). Most of the aircraft profiles range between the surface and 350 hPa. For shorter periods, flights up to 150 hPa are available for the sites Charleston, South Carolina (SCA), SGP, and Cartersville, Georgia (VAA), also covering the UTLS (upper troposphere/lower stratosphere) region.


### 2.7.3. CONTRAIL measurements

The CONTRAIL (Comprehensive Observation Network for Trace gases by AIrLiners) program uses an Automatic air Sampling Equipment (ASE) for flask sampling and Continuous $CO_2$ Measuring Equipment (CME) for in-situ $CO_2$ measurements (Machida et al., 2008; Matsueda et al., 2008). These instruments have been installed on several

Boeing aircraft operated by Japan Airlines (JAL) with regular flights from Japan to Australia, Europe, Asia (East, South and Southeast), Hawaii and North America, providing large spatial data coverage across the globe, particularly in the Northern Hemisphere. The ASE performed flask samplings in the upper troposphere and lower stratosphere (altitude range of ~7–12 km). The CME data are recorded at 10-sec intervals during ascent/descent (~100-meter intervals in altitude) and at 1-min intervals during cruise (~15 km intervals horizontally) as well as in-

flight aircraft positions. To avoid heavy pollution around airports, CME is not operated within ~600 m of the ground surface. The CME has obtained thousands of $CO_2$ vertical profiles over many airports since 2005. The ASE and CME along with NOAA aircraft measurements of $CO_2$ are used to estimate the latitudinal bias in predicted fluxes.





## 2.8 Evaluation Metrics

We calculate correlation coefficients (R), mean bias (MB), and root-mean-square errors (RMSE) to evaluate the

predicted fluxes with aircraft observations. The mean bias and RMSE are defined as

$$MB = \left( \frac{1}{n} \sum_{i=1}^{n} \left( x^o_{aircraft} - x^p_{aircraft} \right)_i \right)$$

$$RMSE = \sqrt{\left( \frac{1}{n} \sum_{i=1}^{n} \left( x^o_{aircraft} - x^p_{aircraft} \right)^2_i \right)}$$


where $x^o_{aircraft}$ is aircraft observation, $x^p_{aircraft}$ is predicted $CO_2$ mole fraction sampled at the i[th] aircraft location,

and n is the number of aircraft observations. The MB infers the magnitude of underestimation/overestimation of

CO2 mixing ratios by the model. The RMSE includes errors (both random and systematic) in the predicted $CO_2$.

The MB and RMSE could be due to uncertainties in predicted fluxes, and model transport. The simulations of $SF_6$

and age of air confirm the low transport error in MIROC4-ACTM (Bisht et al., 2021; Patra et al., 2018). Hence,

the magnitude of biases and RMSE indicates predominantly the accuracy of the predicted fluxes.

## 3. Global-scale land and ocean fluxes

### 3.1 Global distributions

Figure 4 shows the spatial distributions of annual mean $CO_2$ fluxes. While the spatial distributions of a priori

oceanic flux are similar for the JMA and TT09, the terrestrial biosphere fluxes are vastly different. The annually

neutral CASA fluxes show near-zero values for most grids. However, strong sinks are observed over most of the

densely vegetated regions of the globe for the VISIT+GFED fluxes, mainly because VISIT simulation produces

stronger sinks by the ecosystem (Fig. 4a, 4d). Anomalously strong sources are also seen in Fig. 4d due to the fire

emissions estimated by GFED based on the satellite-derived burned area anomaly. The inverse models make

reasonable corrections, though the prior gc3t case includes the net-zero annual flux or case 'gvjf' was a stronger

sink. Consistently predicted fluxes are seen for North America, Europe, Russia, or East Asia for the PFU=ctl case

(middle row). Similarities are slightly less when the PFU is scaled to the GPP of 84 regions of the inverse model

(bottom row). This suggests that the greater PFU is more suitable for the inversion when the region has





observational sites (Fig. 1). In the case of PFU=gpp_v2, the fluxes are not allowed to change much in the boreal regions except for the summer months. However, the gpp_v2 inversion may be preferred over the dry region of Northern Africa, where the control PFU case produces an East (weak source) - West (weak sink) dipole. Performance of the inversions to retrieve the flux distributions over tropical America and tropical Africa are unclear. They show large dependence on the prior flux distributions, possibly due to the lack of observations within

the land regions in our inversion. The main focus of this study is to better understand the total regional emissions and their trends over the past 20 years. The inversion does not revise the fine structures within each of the 84 regions of the inversion by the design of the system, where the regional basis functions assume a fixed pattern for constructing the source-receptor relationships (G-matrix). However, the degree of freedom of our inversions is similar to the gridded inversions when spatial flux correlations of greater than 1000 km are assumed (Peylin et al.,

335    2013).

### 3.2. Global totals.

Figure 5 shows the trends and interannual variability in the global fossil fuel (FF) emissions (used as input for the inverse model), land-biosphere, ocean, and annual atmospheric $CO_2$ growth rate for 16 inversion ensemble

members based on two combinations of land-biosphere and ocean prior fluxes (VISIT and CASA for land-atmosphere, and TT09 and JMA for sea-air) and prior flux/data uncertainties (PFU and MDU). The uncertainty in the ensemble means flux of 16 inversion cases is calculated using ±1σ spread in the time-averaged fluxes for 10 or 5 yr periods in this study. The uncertainties in the predicted fluxes due to different priors are found to be 0.35 PgC yr$^{-1}$ for global land and 0.1 PgC yr$^{-1}$ for the global ocean. The uncertainty due to PFU and MDU is less than 0.15

PgC yr$^{-1}$ for land and ocean carbon uptake for gc3t or gvjf inversions. It indicates that prior flux patterns and trends alter the predicted global land and ocean fluxes. Ensemble means land and ocean fluxes are in excellent agreement with the IPCC "mean" estimates, notably within the 1-σ uncertainty estimated from 16 ensemble member inversions (Table 3). The ensemble spread is much lower (Table 3; MIROC4-ACTM columns), compared to the inversion predicted flux uncertainties, which are in the range of 1.4 and 0.7 PgC yr$^{-1}$ for the global land and ocean,

respectively (vary from low values of 0.8 and 0.5 PgC yr$^{-1}$ for gpp_v2 cases to 1.6 and 0.9 PgC yr$^{-1}$ for the gpp_v4 inversions).





The year-to-year variability in land and ocean carbon sink (Fig. 5b,c) shows considerable agreement across the inversion cases because of the strong constraint provided by atmospheric $CO_2$ measurements at the global scale due to global tracer mass conservation. The year-to-year variability in atmospheric $CO_2$ growth rate is linked to the variability in natural sources and sinks of carbon from land and ocean for given FFC emissions. The observed $CO_2$ growth rate from NOAA (Dlugokencky and Tans, 2020) is compared with the estimated $CO_2$ growth rate (by inversion), defined as the difference between fossil fuel emissions and total sink over land and ocean on an annual basis (Fig. 5d). The NOAA growth rate (ppm yr$^{-1}$) is converted to units PgC yr$^{-1}$ using a conversion factor of 2.13 PgC ppm$^{-1}$ (Raupach et al., 2011). The resulting mean carbon budget imbalance, calculated as the difference between the inversion estimated and the observed $CO_2$ growth rates, is given in the legends (Fig. 5). The year-to-year variability in the global annual total of net $CO_2$ flux is robust across different inversion cases (r=0.97) and with the observed growth rates (r=0.9); however, the global totals over 2001-2020 show slight bias with that observed. Compared to the observed $CO_2$ growth rate, the inversion shows systematic positive (range 0.1-0.3) and negative (range 0.0 - (-0.2)) imbalances for gc3t and gvjf inversions, respectively. The ensemble mean of 16 inversions ("ensm") agrees well with the observed growth rate within the uncertainty of the predicted fluxes over the 20 years (Fig. 5d).

Inversions suggest that both the terrestrial land and ocean sinks increased during our analysis period 2000 to 2020. The ensemble means of terrestrial land $CO_2$ sink increased from -2.31±0.21 PgC yr$^{-1}$ in the 2000s (2000-2009) to -2.9±0.26 PgC yr$^{-1}$ in 2010s (2010-2019) with significant interannual variations up to 2 PgC yr$^{-1}$. The interannual variability in land $CO_2$ flux is predominately associated with the response of the terrestrial biosphere to El Niño and its cold counterpart La Niña (collectively known as El Niño-Southern-Oscillation or ENSO: colour shading in Fig. 5; +ve for El Niño and -ve for La Niña) )-induced changes in the weather pattern. In 2015-2016, and to a lesser extent in 2010, El Niño conditions reduced carbon uptake by the land ecosystems in the tropics (e.g., Baker et al., 2006; Bousquet et al., 2000; Patra et al., 2005b). The average ocean sink intensified from -1.46±0.10 PgC yr$^{-1}$ in the 2000s to -1.62±0.17 PgC yr$^{-1}$ in the 2010s, with interannual variations of the order of a few tenths of PgC yr$^{-1}$ ; generally showing an increased ocean sink during strong El Niño events (e.g., during 2015-2016) (Chatterjee et al., 2017; Feely et al., 1999; Patra et al., 2005a). The global ocean biogeochemistry model (GOBM) products, and those based on pCO$_2$ measurements, also show similar decadal variability patterns (DeVries et al., 2019; Hauck et al., 2020).



This gradual increase in the ocean $CO_2$ sink is caused by increasing $pCO_2$ between the marine air and sea-surface water. The strong increase in the net sink by the terrestrial biosphere until about 2012 is sometimes attributed to the bias in FFC emissions from China (Saeki and Patra, 2017), and the gradual sink increase can be attributed to

$CO_2$ fertilisation or water-use efficiency as more carbon is available for assimilation by photosynthesis (Keeling et al., 2017; Kondo et al., 2018). In addition, the land $CO_2$ uptake efficiency in the period of this analysis could partly be controlled by a decadal shift in frequency of natural climate variability, such as ENSO. In the 2000s, no extreme El Niño conditions were observed, resulting in suppressed fire emissions and lowered drought occurrences, while the opposite conditions prevailed in the 2010s (intense El Niño in 2010, 2015-2016, 2019-2020), which is likely to

reduce net uptake by the land ecosystems (Patra et al., 2005b). Fire emissions with an estimated peak-to-trough change of about 0.5 PgC yr$^{-1}$ (Table 1) alone cannot explain the large changes in land sinks of the order of 2-3 PgC yr$^{-1}$.

## 4. Sub-continental scale land and ocean fluxes

### 4.1 Interannual variability and uncertainties

Here, we present interannual variability in a priori and predicted carbon fluxes over 15 lands (Fig. 6) and 11 ocean regions (Fig. 7). The uncertainty in the predicted carbon fluxes is estimated as 1-$\sigma$ spread among the 16 inversion cases. Significant differences are seen in between a priori (VISIT) and predicted land fluxes in the high carbon sink regions, e.g., over Russia (-0.76 PgC yr$^{-1}$), followed by East Asia (-0.55 PgC yr$^{-1}$), and Europe (-0.54 PgC yr$^{-1}$),

while the inversion suggests the more substantial uptakes over Temperate North America (-0.59±0.14 PgC yr$^{-1}$), followed by East Asia (-0.49±0.09 PgC yr$^{-1}$), Boreal North America (-0.38±0.1 PgC yr$^{-1}$), and Russia (-0.35±0.05 PgC yr$^{-1}$) for the study period.

Large spread in the predicted fluxes over Temperate North America, Temperate South America, and Europe

indicate that the estimated flux is more sensitive to the selection of PFU and MDU when there are measurement sites within the region or in the neighborhoods. The carbon fluxes over tropical (Tropical America, Brazil, South Asia, and Southeast Asia) and extratropical (Temperate South America, Central Africa, Southern Africa, and Oceania) land regions are found relatively less certain; the ensemble of inversions splits into weak source or sink group based on the land priors. For example, inversions using the VISIT flux show a net source signal over all the

three African regions, while those using the CASA flux exhibit a net sink. Similarly, for South Asia, all inversion





cases based on the CASA prior show a mild carbon sink, while VISIT-based inversion cases show a net sink of -0.18 ± 0.11 PgC yr$^{-1}$. VISIT prior consists of very strong sinks over all 3 South America regions (Fig. 6; Table S2). These regions of Africa, South America, and South Asia are weakly constrained in the inversions due to the limited observations representing these regions. However, for most of these regions lacking in observations, the VISIT and

CASA-based inversions are moving toward a common flux value, i.e., the range of the two prior fluxes are usually much greater than the 16-inversion ensemble spread.

The predicted land carbon sink over East Asia tends to increase, consistent with a rapid increase in FFC emissions for 2001-2009. The rapid increase in $CO_2$ emissions from FFC could impose an artificial trend in the terrestrial

land flux estimate for the East Asia region (Saeki and Patra, 2017). Because the atmospheric data constrain the total net surface flux, the rapid increase in fossil fuel emissions is required to be compensated by increasing the natural land uptake of similar magnitude through inversion. The South Asian uptake remains almost constant for the study period. For West Asia, both sets of inversion show the land $CO_2$ flux fluctuates around zero with slight interannual variation, indicating a stable trend of land flux changes and a small contribution to uptake of all the

tropical regions. The annual trend in Southeast Asian carbon fluxes is overwhelmed by large interannual variability, driven mainly by ENSO-induced fire emission variability (Patra et al., 2005b; van der Werf et al., 2017). Over the African continent, Central Africa shows the highest interannual variability, mainly due to biomass burning emissions under the influence of ENSO. The $CO_2$ flux anomalies in the case of gvjf inversions supports the fire emission anomalies taken from GFEDv4s as included in the prior, and what is more impressive for us is the ability

of the gc3t inversions to produce similar phase and magnitude of the flux variabilities while the prior flux consisted no interannual variability (even for these relatively unconstrained regions).

Figure 7 shows good agreement among the predicted fluxes for the 11 ocean regions and the decadal flux variability, which are derived from TT09 prior flux with no interannual variability and JMA flux including interannual

variability. The inversion results show substantial changes in the estimated interannual variability due to PFU and MDU. The Northern Ocean shows a significant spread from the mean (-0.13 ± 0.14 PgC yr$^{-1}$) for higher MDU, particularly for 2011-2015 (-0.01±0.06 PgC yr$^{-1}$) gvjf inversions. In opposite phase a similar spread is also found for the neighboring Boreal North America, Europe, and Russia land fluxes. Because the flux variabilities over land regions are larger than ocean regions (by about a factor of 2), the Northern Ocean fluxes can be perturbed by





relatively unnoticeable anomalies in the neighbouring land fluxes. It is indicative that an analysis of fluxes for one region may be challenging to interpret when isolated from the rest of the world in an inversion of long-lived species because of the large-scale mixing by transport.

A significant difference in the priors over the South Atlantic is observed; the JMA fluxes suggest two-fold $CO_2$ uptake (0.33 PgC yr$^{-1}$) than in the TT09 (Table S2). Inversion largely follows their priors over this region, which

is observationally unconstrained by sites within the region. Both the prior and predicted fluxes show that the East Pacific as a significant source of $CO_2$ to the atmosphere, caused by the upwelling of deep ocean water (that brings $CO_2$-rich water from ocean interior to surface) of the west coast of the South American continent. A small net outgassing signal is also known to have over the Tropical Indian and Tropical Atlantic Oceans. Because of tighter constrain by relatively extensive coverage of observation sites, all 16 predicted fluxes converged on consistent

Southern Ocean decadal trends and interannual variability; while, we have not accounted for interannual variability or trends in the TT09 prior. The Southern Ocean $CO_2$ sink intensity shows considerable variability from interannual to decadal timescales, and sink stabilisation after 2010 may have been caused by a regional shift in sea level pressure and surface winds (Keppler and Landschützer, 2019).

**4.2 Interannual variations in regional $CO_2$ fluxes**

To examine the regional pattern of anomalies in the land and ocean $CO_2$ sink, following Patra et al. (2005b, 2005a), we calculated the monthly anomalies by subtracting a long-term mean seasonal cycle from the monthly emissions from 2001-2020 (Fig. 8). Thus, the time series contains the interannual variability (IAV) and long-term trends for the analysis period. We find that the anomalies from both inversion cases are consistent over most regions; however

the magnitude differs. Despite the absence of IAV in the "gc3t" prior fluxes, the consistent interannual variability suggests that inversion is robust in constraining the IAV in carbon fluxes. The correlations between "gc3t" and "gvjf" inversion over land are greater than 0.7, which are statistically significant at p<0.0001. The low correlation (less than 0.3) between "gc3t" inversion and "gvjf" prior can be inferred as only some of the interannual variabilities derived by inversions were present in the gvjf prior, and the interannual flux variability for gvjf inversions are

significantly different from those present in a priori. Large-scale cyclic patterns of climate anomalies such as ENSO account for a large part of climate variability on interannual to sub-decadal timescales $CO_2$ flux variabilities.





Climate anomalies are associated with changes in temperature distributions and large-scale circulations of the ocean water and the atmosphere.

The $CO_2$ flux anomalies in the tropical regions are strongly correlated with the ENSO index, while temperate and boreal regions are weakly correlated, as expected from the areas of ENSO influences (Table S3). In the northern high latitude regions of Europe, America, and Asia, a negligible correlation was found between $CO_2$ flux anomalies and the ENSO index. It is because, over these regions, the North Atlantic Oscillation (NAO)/Arctic Oscillation (AO) or Pacific Decadal Oscillation (PDO) are the dominant climate factors for the CO2 flux anomaly, likely
through the temperature variation (Patra et al., 2005). The warmer weather in these regions may lead to a negative $CO_2$ flux anomaly (Russia) since that condition stretches the growing season length (Dye and Tucker, 2003). The correlation coefficient between $CO_2$ flux anomalies for different aggregated land and ocean regions is given in Table S3.

In 2015, a strong El Niño induced severe drought and biomass burning in Equatorial Asia. It was one of the most significant El Niño events after the well-known major El Niño in 1997/1998 (L'Heureux et al., 2017; Santoso et al., 2017). Patra et al. (2005b) estimated that a massive amount of carbon (~5.5 PgC yr$^{-1}$) released into the atmosphere in 1997/1998, largely contributed from tropical regions in Asia, South America, and Africa. Figure 7 shows that during the Extreme El Niño period in 2015-2016, large carbon was also released from Tropical America
and Southeast Asia, followed by Central Africa and Brazil. However, the timing and strength of the peaks are different. The fire emission peak (for "gvjf" case) appeared in August 2015 over Southeast Asia (1.34 PgC yr$^{-1}$), and in January 2016 over Tropical America (1.12 PgC yr$^{-1}$). The "gc3t" inversion suggests a peak in October 2015 for Southeast Asia (0.42 PgC yr$^{-1}$) and in January 2016 for Tropical America (0.82 PgC yr$^{-1}$). Niwa et al. (2021) showed that regular sampling of aircraft $CO_2$ measurement under CONTRAIL project has enormous potential for
capturing the footprint of biomass burning. By advanced inverse analysis, they estimated equatorial Southeast Asia emission at 0.27 PgC yr$^{-1}$ during September-October 2015 (note that northern Southeast Asia is not included). Over the Oceania regions, 'gvjf' inversion shows large interannual variability. The inversion suggests emission peaks in 2019-2020 over Oceania, consistent with the substantial $CO_2$ released from the fire in the atmosphere during the 2019-2020 summer season over Australia (van der Velde et al., 2021). They concluded that the $CO_2$
emission from these fires was more than thrice the estimate derived from the long-term mean of $CO_2$ uptake over this region and broadly consistent with estimates based on the GFED fire emissions. van der Velde et al. (2021)



estimated $CO_2$ flux anomaly in the range 0.14–0.24 PgC from November 2019 to January 2020, which are lower than our estimation of 0.7 PgC.

$CO_2$ flux variabilities over East Pacific, West Pacific, and South Pacific show negligible correlations with ENSO, while both JMA based prior and posterior show high correlation (Table S3). Over the North Pacific, both inversion ensembles show an insignificant correlation with ENSO. The oceanographic observations indicate that sea surface temperature and $pCO_2$ in the equatorial warm pool areas (5°N–5°S, west of the dateline) are not sensitive to El Niño conditions (Takahashi et al., 2003). $CO_2$ flux anomalies are estimated positive for South Pacific and negative

for East Pacific during the 2015-2016 El Niño event. On the contrary, observations show that sea-to-air $CO_2$ flux is suppressed during the intense El Niño event by warm low-$CO_2$ surface waters from the west. The atmospheric observations are limited over these regions. Thus, we consider the anomalies during the intense El Niño period 2015-2016 are likely to be an artifact because of the lack of observational constraints. The interannual variability over tropical Atlantic, tropical Indian, south Indian is low. Though the IAV is low, we find a significant correlation

for the tropical Atlantic for both predicted fluxes, in contrast to a negligible correlation between prior and ENSO.

## 4.2 Mean seasonal cycle

The net $CO_2$ uptake in the moist/warm growing season is partially compensated by net $CO_2$ release in the dry/cold non-growing season. However, the magnitude of seasonal compensations differed significantly between regions.

The compensation drives the net regional strength of $CO_2$ uptake. For example, the maximum uptake in the growing season is shown by Russia (~10 PgC yr$^{-1}$) followed by Boreal North America (~5 PgC yr$^{-1}$), and temperate North America (4 PgC yr$^{-1}$) (Fig. 9). However, the strength of net sink is almost twice over Temperate North America (-0.59±0.14 PgC yr$^{-1}$) than in Russia and Boreal North America, because of the release of $CO_2$ in the non-growing season over Russia and Boreal North America are greater than Temperate North America. Thus, the seasonal cycles

in global and regional emissions are essential for understanding the drivers of $CO_2$ changes in the atmosphere.

Seasonal cycle amplitude for CASA prior flux for land total is 33.6 PgC yr$^{-1}$, and VISIT is 23.8 PgC yr$^{-1}$, and the peak of the growing season (when the net flux is most negative) occurred in July, that is one month after the VISIT (Fig. 9, top-left panel). The seasonal phase of "gc3t" inversion ensemble cases are in close agreement with the

CASA prior, but inversions diverge for the net $CO_2$ uptake and release in the growing and dormant seasons, respectively. Our inverse analysis indicates more considerable $CO_2$ uptake (-26.2±2.1 PgC; June - July) and net



$CO_2$ release (4.1±2.0 PgC; January to April) than CASA-based terrestrial land flux (Fig. 9a). Contrary to the CASA prior, inversions using VISIT increase the sinks to June-July compared to a priori. Inversion fluxes using CASA and VISIT show very high consistency for the total land $CO_2$ flux seasonality. Overall, the averaged "gc3t"

and "gvjf" show good agreement (r=0.98) after inversion as compared to prior (r= 0.8). Inversions do not achieve similar consistency for the total global ocean fluxes for both TT09 and JMA a priori fluxes. The correlations reduced from 0.95 to 0.58 after the inversion.

Northern land drives the global land seasonality, with a close agreement regarding the magnitude and phasing of

the growing season and dormant season fluxes. Seasonality for the tropical land (South Asia, Tropical America, Central Africa, Southeast Asia) is smaller than the northern land regions (Boreal North America, Temperate North America, Russia), with the inversion suggesting maximum uptake around June-July over northern land and July-September over tropical land. The a priori and predicted fluxes are more consistent for the extratropical land regions than their tropical counterparts. This is because the temperate biosphere are better simulated by the ecosystem

models, such as VISIT or CASA, by taking in to account the temperature and light effects, while the tropical ecosystems are more often limited by water availability or suffers from extreme heat, e.g., the monsoon driven South Asia (Patra et al., 2013).

Over Tropical America, CASA shows maximum carbon uptake (flux=-2.0 PgC yr$^{-1}$) in August and an extended-

release period in dormant season from January to April (flux=1.1 PgC yr$^{-1}$), while VISIT shows a relatively small seasonal variation. However, the inversions derive a consistent seasonal phase and amplitude, although the region does not include any measurement site. Thus, the neighbourhoods' observations are helping us to capture the signal from this region, which is probably enabled by a good transport simulation by MIROC4-ACTM. Similar improved consistency in predicted seasonal cycle phase and amplitude, relative to the a priori, are also obtained for many

other regions (Fig. 9a,b,c,f,g,k,n). Nevertheless, the a priori models play significant roles in estimations of $CO_2$ flux seasonality for South Asia (peak uptake flux -0.43 PgC yr$^{-1}$ in August-September for VISIT, and -1 PgC yr$^{-1}$ for CASA in October-November), East Asia (CASA peak uptake in August, VISIT peak uptake in April).

The global ocean prior fluxes show the weak uptake of $CO_2$ during July-September (average uptake flux of -0.53±0.09 PgC yr$^{-1}$ for TT09 and -1.12±0.14 PgC yr$^{-1}$ for JMA). After September onwards, a sharp increase in uptake occurs, with a maximum uptake of 2.43 PgC yr$^{-1}$ for TT09 and 2.91 PgC yr$^{-1}$ for JMA fluxes in December



(Fig. 9, left-3$^{rd}$ row from bottom). Over the Northern Ocean, the "gvjf" inversion cases tend to show the large $CO_2$ release during May-October as the MDU increases. We believe the broader uptake seasons for Boreal North America, Europe, Russia, leading to stronger early summer land uptake in the case of VISIT a priori caused positive $CO_2$ flux seasonality for the Northern Ocean. Even for the gc3t inversion case we find the peak in seasonal cycle in the summer season, when the oceanic biosphere activity is at its peak (Goto et al., 2017; Yasunaka et al., 2018). It is not easy for us to explain the mechanism for the Northern Ocean to be a weaker sink in summer than in winter. Further studies are needed to identify the role of ice-covered areas (close to zero $CO_2$ flux) on the seasonal cycle. Note here that the oceanic basis functions in the polar ocean regions use a climatological sea-ice cover map, and fluxes are not revised over the sea-ice-covered areas.

Overall all the land regions, except South Asia, Southeast Asia, Central Africa, and Oceania, show very good agreement between averaged "gc3t" and "gvjf" cases after inversion (r = 0.63 – 0.97), as compared to priors (r = - 0.48 – 0.90). A good agreement over the Northern Ocean, West Pacific, and Tropical Atlantic Ocean seasonal cycle is observed after the inversion (Table S3).

## 5. Regional fluxes and uncertainty estimate

Different regions across the globe emit different amounts of $CO_2$ from FFC emissions, which is one of the main discussion points in the emission reduction policymaking, say under the Kyoto Protocol (1997) and Paris Agreement (2015) for limiting global warming below a certain level. The recent IPCC AR6 of Working Group I suggested that the global total $CO_2$ emissions from FFC have to be removed gradually on the net annual basis by 2050 to sustainably limit global warming below 1.5$^o$C (Canadell et al., 2021). Because the elimination of FFC usage is challenging to envisage in the coming three decades, pathways for reducing FFC emissions are being explored. Carbon capture and storage and other technological managements are considered alongside the nature-based solutions. The land and ocean have been helping to remove more than 50% of the FFC emissions in the past decades. The ongoing natural sinks of $CO_2$ and their maintenance/enhancement constitute the major theme of the nature-based solutions. Thus, it is imperative to understand global/regional carbon fluxes for developing national and international policy to reduce net $CO_2$ emissions.

Figure 10 and Supplementary Figures S3 and S4 show regional $CO_2$ fluxes and flux uncertainties from the 16-member ensemble inversions for 15 land and 11 ocean regions and global land and ocean at 5-yr intervals for the past two decades. The global flux uncertainties are found smaller than the regional flux uncertainties because the



former is constrained strongly by the atmospheric $CO_2$ growth rate for given FFC emissions. Flux estimates for all
the land regions remain quite uncertain, as seen from the 75 percentile of the 16-inversion ensemble (error bars) at
about 0.3 PgC yr$^{-1}$ for the land regions and typically less than 0.2 PgC yr$^{-1}$ for the ocean regions. The fluxes at 25
percentile show slightly reduced uncertainties – a large reduction is not seen compared to the 75 percentiles fluxes
because the two a priori models often formed two different sets of $CO_2$ flux values (ref. Fig. 6 and 7). However, it
has to be noted that each of the 15 land analysis regions containing at least three inversion regions will have more
than 4 PgC yr$^{-1}$ predicted flux uncertainty, as the reduction from prior flux uncertainties was meagre by inversion
(Fig. 3). Thus, by employing the 16-inversion ensemble approach, we could obtain flux uncertainties that are an
order of magnitude smaller and often less than the regional fluxes themselves. The mean/median fluxes are
consistent for the ensemble inversions and represent the true state of $CO_2$ flux estimation for the MIROC-ACTM
and 50 sites used in the inversion.


Global land sink increased by ~63% from 2001-2005 to 2016-2020 with an uncertainty range of ~6-12%. The
highest increase (about 73 %) in the land sink is observed from 2001-2005 to 2011-2015, while a decrease (~11%)
is observed from 2011-2015 to 2016-2020. The northern extratropical land accounted for ~80% of the global land
sink, followed by tropics (~13%) and southern extratropic (~7%) (Fig. S3). The ocean carbon sink shows a gradual
increase (by ~30%) from 2001-2005 to 2016-2020. The southern extratropic represents about 85% of global ocean
sink for 2001-2020, followed by the northern extratropic (~60%). The tropical ocean regions act as a net source of
carbon emissions, representing 45% of global ocean carbon sink (Fig. S4). One of the most intriguing features is
that the 5-year mean fluxes for the ocean have increased gradually, as expected from the increase in pCO$_2$ partial
pressure difference due to increased loading of FFC emissions, but the land flux increases only during 2001-2005
and 2006-2010. This step increase in flux can be related to the biased FFC emissions from China, affecting the
natural/managed land flux estimation by inversion (Saeki and Patra, 2017).

Amazonia in Brazil, host the Earth's largest tropical forest, hence an important region of carbon sink. Our study
shows a slight decrease in carbon sink over this region from 2011 to 2020. A recent study based on aircraft
measurements (Gatti et al., 2021) also suggests a decline in carbon sink from 2010 to 2018 over Amazonia due to
factors such as deforestation and climate change. A very high correlation is also seen for the interannual and decadal
variations in $CO_2$ fluxes (Fig. 6d, Fig. 10) and the Brazilian Amazon deforestation rate, which showed a strong and
systematic decline from the period 2002-2004 to 2012-2014, and a steady increase afterward (Silva Junior et al.,
2021).




Our results show that Africa is a small sink of carbon on an annual scale, agreeing with the RECCAP -1 estimation for 2000-2009 (Table. 4). At the subregional level, northern Africa shows a small sink, while Central and Southern Africa show a minor source for the same period. Central Africa turned from a small source in 2000-2009 to sink in 2010-2019, while the carbon flux behaviours remain the same for Northern Africa and Africa at a higher magnitude.

Though Central Africa is the main carbon sink region over the African continent because of its evergreen tropical forest, prolonged dry season due to El Nino during 2001-2005 could turn it into a net source region for the 2000s. The average annual mean fluxes over East Asia for the 2000s are remarkably consistent with the RECCAP estimates, based on the average estimate from inventory, bottom-up, and inversion fluxes (Piao et al., 2012). We have observed less sink for Russia than RECCAP best estimate ((Dolman et al., 2012). Other regional flux also

agrees well with RECCAP estimates, although the period and regional boundaries of the RECCAP assessment do not match precisely (Table 4).

Our analysis suggests that the most prominent land carbon sink in the Northern Hemisphere is unanimously located in temperate North America ($-0.59\pm0.14$ PgC yr$^{-1}$), followed by East Asia ($-0.49\pm0.09$ PgC yr$^{-1}$), boreal North

America ($-0.38\pm0.10$ PgC yr$^{-1}$), and Russia ($-0.35\pm0.05$ PgC yr$^{-1}$) for 2001-2020, and they account for 70% of the total global CO2 uptake by land biosphere. Overall, our results suggest about 40% of temperate North America's (1.49 PgC yr$^{-1}$), 17-19% of East Asia's (2.74 PgC yr$^{-1}$), 200% of boreal North America's (0.19 PgC yr$^{-1}$), and 80% of Russia's (0.44 PgC yr$^{-1}$) $CO_2$ emissions from FFC are offset by carbon accumulation in their terrestrial ecosystems for 2001-2020. Overall, no area shows net carbon source from the land biosphere for recent decades

(2010-2019). Further, the inversion suggests the most substantial oceanic $CO_2$ uptake in the north pacific with a mean flux of $-0.55\pm0.05$ PgC yr$^{-1}$. However, the most considerable $CO_2$ uptake rate is observed in the Southern Ocean region; the $CO_2$ flux increased from $-0.12\pm0.07$ PgC yr$^{-1}$ in 2001-2009 to $-0.33\pm0.06$ PgC yr$^{-1}$ in 2010-2019. Southern Ocean $CO_2$ flux for 2010-2019 agrees well with a recent assessment of $-0.53 \pm 0.23$ PgC yr$^{-1}$ (net uptake) in the region south of 45°S during 2009–2018 (Long et al., 2021).


## 4. Validation of $CO_2$ fluxes using aircraft data

We evaluate the quality of inversion flux estimates by comparing $CO_2$ simulations with independent observations (observations that are not used in the inversions due to lack of long measurement time series record). The $CO_2$ simulations are derived from three sets of prescribed fluxes: "gc3t", "gvjf", and "ensm". The observations in the

lower troposphere (from surface to ~2 km) are more sensitive to regional fluxes. Hence, we compare the simulated



$CO_2$ with those measured by HIPPO and Atom airborne campaigns in the lower troposphere. Figure 11 shows comparisons over the transects from high northern (~80°N) to high southern latitudes (~70°S) at the location and time of HIPPO and Atom airborne campaigns, spanning all four seasons. HIPPO shows the lower $CO_2$ over 30-80°N than 0-30°N for May-July due to large uptake in high northern latitudes; however, the values are slightly
higher than Southern Hemisphere (Fig. 11, Fig. S5). ATom shows a lower concentration over 30-80°N than the rest of latitudes during July-August (Fig. 11, Fig. S6). All the model cases capture the meridional gradient, slope, and other features well at RMSEs less than 1.5 ppm, mean bias in range -0.5 – (-0.3) ppm, and correlation greater than 0.8.

The "ensm" inversion shows the lowest mean bias and RMSE than the other two predicted simulations over most aircraft campaigns (Fig. 11, statistics on each panel). The comparison also indicates latitudinal and seasonal dependent accuracy in the predicted fluxes. The observations (ATom and HIPPO) show a high bias during boreal summer throughout the troposphere over the northern high latitude regions (>50°N), implying possible seasonally dependent errors in posterior fluxes over these latitude regions. Most of the aircraft data over these latitude bands
are available over the continental regions, and this comparison suggests a higher sink than the estimated sink by inversion. Based on this comparison, the simulations from the ensemble mean of 16 inversion cases ("ensm") is well-suited (mean bias is found to be minimum) for quantifying the global land and ocean carbon sink on the timescale of annual mean and its decadal trend.

Further, all available aircraft profiles, measured on a campaign basis or regular intervals, are also used to evaluate the predicted fluxes (Table S4). Compared to NOAA vertical profiles of $CO_2$, model simulations agree well in the free troposphere (defined here between 2 and 8 km), with an average bias (averaged over 2000–2020) close to zero (Figure S7, top panel: bias as a function of altitude, averaged over all sites). Inversion underestimates (~1 ppm; Figure S7, top panel) the observations within the boundary layer (between the surface and 2 km); however, the
RMSE is higher (~1 ppm) compared to that of the free troposphere.  It could be because many of the NOAA aircraft profiles are over the U.S. (see the map inset in the middle row, left panel of Fig. S7), close to regional $CO_2$ sources.

Following the GCP budget evaluation method (Friedlingstein et al., 2020), Figure 12 shows the mean bias over four latitudinal bands for three periods. The mean bias is calculated separately from all available aircraft data in
obspack_co2_1_GLOBALVIEWplus_v6.1 (Fig 12a-c) and 50 observation sites (Fig. 12d-f) used in the inversion (shown in Fig. 1). The biases show the dependency on latitude and are different for each inverse model case and





provide information on biases in the surface fluxes (Gaubert et al., 2019; Houweling et al., 2015). All the three predicted $CO_2$ concentrations show the smallest bias (less than 0.1 ppmv) over the high southern latitudinal band for 2001-2010. However, for 2011-2020, predicted $CO_2$ concentrations show large positive ($0.47 \pm 0.47\ ppm$)

and negative ($-0.23 \pm 0.47\ ppm$) bias for gc3t and gvjf predicted fluxes. The positive and negative biases for gc3t and gvjf inversions, respectively, are also consistent with the surface sites, which are arising due to the biases in simulated $CO_2$ growth rates (an overestimated growth rate for gc3t for the +ve budget imbalance, Fig. 5; opposite being the case for gvjf inversion). The mean biases turned from negative to large positive by moving from 0-45ºN averages to 45-90ºN averages. In contrast to aircraft observations, the surface sites show a large negative bias over

45-90ºN.

The large model-observation bias for the simulations of predicted fluxes, compared to the predicted data (Fig. 12d,e, f; solid and broken lines, respectively; using the 50 sites used in inversion), are likely to be caused by the propagation of biases in regional fluxes due to the sparse observational coverage in the tropical region. In the case

of predicted data, the inversion fits the observation well due to minimisation of prior model-observation differences, but when the simulations are run using predicted fluxes, the (small) systematic biases produce a (large) cumulative effect over the model integration period. It is also interesting to note that the meridional gradients in biases for independent aircraft observations (Fig. 12a,b,c) and sites used in inversion (Fig. 12d,e,f) show opposite phases, i.e., most negative and most positive at 25ºN, respectively. We speculate that MIROC4-ACTM produces stronger sinks

in the high northern latitudes (negative model-observation bias at surface sites over 75ºN or HIPPO/ATOM latitude-altitude plots in Fig. S5, S6), which can arise from the model's inability to simulate the sites over the land because of the coarse horizontal resolution. Thus, resulting in a weaker sink or a stronger source in the northern tropics and subtropical (25ºN) regions, respectively. The tropical source is then transported to the mid-high latitudes, which is captured by the aircraft observations, as a positively biased concentration. This experience

suggests a need for new forward model simulations using inversion fluxes, not the optimised atmospheric $CO_2$ fields during data assimilation, should be used for evaluating inversion fluxes with the help of independent observations.

Further, the fluxes should be evaluated by independent metric, i.e., not by using the assimilated $CO_2$ field for the

independent $CO_2$ measurement, which is a "perfect" transport model condition (ref. schematic diagram Fig. S10). This has been dealt with in great detail in the TransCom-CH4 experiment. For example, when a model produced a stronger northern-to-southern hemisphere (NH-SH) gradient for a tracer of inter-hemispheric (IH) transport,



sulphur hexafluoride ($SF_6$), the model also produced a stronger NH-SH gradient for methane ($CH_4$). This implied that models with a stronger NH-SH $SF_6$ gradient would require a smaller source of $CH_4$ in the NH compared to the model producing a weaker NH-SH $SF_6$ gradient. For $CO_2$ inversions, however, the effect of NH-SH transport bias may not produce systematic flux biases because $CO_2$ sources and sinks are much greater compared to the net fluxes at any given time intervals. Complex interactions between vertical and horizontal transport may have mixed impact on flux inversions (Denning et al., 1996; Krol et al., 2019). An objective analysis of inverted $CO_2$ flux requires to be independent from the transport model, and that can only be developed in a multi-model experimental framework, e.g., TransCom.

## 5. Conclusions

The terrestrial biosphere and ocean absorb about 53% of the total anthropogenic carbon fluxes every year. The variability in these fluxes significantly affects the year-to-year variability of the carbon dioxide ($CO_2$) accumulation rate in the atmosphere. We estimated global and regional sources and sinks of carbon across the earth's surface, using observations and simulations of atmospheric $CO_2$. We tested the relative role of prior flux uncertainty (PFU) and measurement data uncertainty (MDU) on the land and ocean's global/regional carbon flux estimates. For this, we use a single transport model (JAMSTEC's MIROC4-ACTM) to estimate the global/regional carbon flux and associated uncertainty based on different prior fluxes (two land biosphere fluxes (CASA and VISIT+Fire) at 3-hourly intervals with distinct global total $CO_2$ sink strengths and two sea-air exchange fluxes (Takahashi and JMA)), choices of prior flux uncertainties (PFU: four cases) and representation of measurement data uncertainties (MDU: two cases). Based on different combinations of priors, PFU, MDU, we run 16 ensembles of inversion cases using $CO_2$ measurements from a set of 50 sites spreading over the globe, and inversion results are analysed for 2001-2020. The 16 ensemble members are averaged, and the resultant ("ensm") is treated as the best estimate (i.e., a measure of central tendency) flux. The spread between the ensemble members provides us a reasonable measure of the inversion estimated flux uncertainty but lacks the quantification of transport model uncertainty.

Although each inversion used common observation sites, there is still considerable uncertainty in the estimated fluxes due to the prior fluxes and uncertainties (PFU and MDU). At a global scale, the uncertainty in the predicted fluxes due to prior flux is relatively larger than that of PFU and MDU. However, at a regional scale, uncertainties due to priors and PFU & MDU are primarily comparable and drove inter-inversion disparity. Lack of constraints makes the tropical (Tropical America, South Asia, and Southeast Asia) and extratropical (Southern Africa, Oceania) land regions highly uncertain. The ensemble of inversions splits into a "near-neutral" group and a "strong-



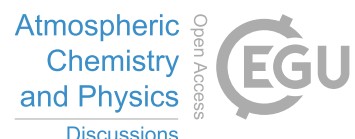

source" group based on the priors. Overall, the prior uncertainties have a negligible effect on the global mean land
and ocean sink but are significant at the regional scale.

The interannual variability in the land fluxes (driven mainly by ENSO) is much more considerable than the ocean
and tends to show greater consistency across the 16 inversion cases. The land flux seasonality is prominent in the
Northern Hemisphere, with good agreement across the inversion ensemble cases. The ocean seasonality is slight
and, in less agreement, relative to the magnitude of the seasonality of land fluxes. We comprehensively evaluated
predicted fluxes by extensively comparing the simulated posterior $CO_2$ to independent $CO_2$ observations from
several aircraft measurements by NOAA, CONTRAIL, HIPPO, ATOM, and the NOAA MBL reference sites. The
evaluation suggests that the ensemble mean of 16 inversion cases ("ensm") is well-suited (mean bias is found
minimum) for quantifying the global land and ocean carbon sink. Thus, we used the "ensm" case (best estimate)
for estimating the carbon fluxes, and associated uncertainties.

In summary, our best estimate suggests that 22-33% (16-18%) of global $CO_2$ emissions from FFC have been offset
by global land (ocean) carbon flux over the period from 2001-2020. The land and ocean sink partitioning are
estimated to be -2.27±0.2 (±1σ uncertainty on mean) and -1.46±0.09 PgC yr$^{-1}$, respectively, for the period 2001-
2009, and -2.85±0.25 and -1.63±0.17 PgC yr$^{-1}$, respectively, for the period 2010-2019 (without riverine export
correction), which are in excellent agreement with the IPCC AR6 values (Table 3).  The decadal mean values for
the RECCAP are also in good agreement for North America, South America, East Asia, South Asia, Oceania (Table
4). Please note that the region boundaries and period of evaluations do not match perfectly between RECCAP-1
and our inversion for most regions.


Our analysis suggests that the most prominent land carbon sink in the Northern Hemisphere is unanimously located
in temperate North America (-0.59±0.14 PgC yr$^{-1}$), followed by East Asia (0.49±0.09 PgC yr$^{-1}$), boreal North
America (-0.38±0.10 PgC yr$^{-1}$), and Russia (-0.35±0.05 PgC yr$^{-1}$) for 2001-2020. Overall, our results suggest about
40% of temperate North America's (1.49 PgC yr$^{-1}$), 17-19% of East Asia's (2.74 PgC yr$^{-1}$), 200% of boreal North
America's (0.19 PgC yr$^{-1}$), and 80% of Russia's (0.44 PgC yr$^{-1}$) $CO_2$ emissions from FFC are offset by carbon
accumulation in their terrestrial ecosystems for 2001-2020. Further, the inversion suggests the most substantial
oceanic $CO_2$ uptake in the north pacific with a mean flux of -0.55±0.05 PgC yr$^{-1}$. However, the most considerable
$CO_2$ uptake rate is observed in the Southern Ocean region; the $CO_2$ uptake increased from -0.12±0.07 PgC yr$^{-1}$ in
2001-2009 to -0.33±0.06 PgC yr$^{-1}$ in 2010-2019. Overall, no area shows net carbon source from the land biosphere

for recent decades (2010-2019). The detailed study of underlying mechanisms of carbon sink over the studied regions gives scope for future work in continuing this study.

There is no doubt that this set of results is unique because they close the year-to-year budget of decadal $CO_2$ changes in the atmosphere. The bottom-up inventory or other modelling system still has limitations in closing year-to-year

budgets. Thus, our tools are robust enough to test our current knowledge of exchange between all carbon pools due to combined processes. This dataset can be used to evaluate biogeochemical models and support the monitoring of regional contributions to changes in atmospheric $CO_2$.

**Code availability**: The $CO_2$ inversion code is based on that was provided by TransCom (Last access: 01/2001), and the revised inversion code is available from https://github.com/prabirp/co2l2r84.

**Data availability**: All flux results are available by requesting the lead/corresponding authors (due to the lack of online storage space). The subset of data (2 inversion cases) is available from the Zenodo (DOI: 10.5281/zenodo.5776197 and DOI:

10.5281/zenodo.5776212). All $CO_2$ observation data used in inversion are available from ObsPack-CO2 (https://gml.noaa.gov/ccgg/obspack/) or the WMO World Data Centre for Greenhouse Gases (WDCGG) (https://gaw.kishou.go.jp/).

**Supplement link**: The article contains Supplementary information and a link to the supplement will be included by Copernicus.

**Author contribution**: NC and PKP designed the experiments, carried them out, performed analysis. MT developed the model codes, DG and SM conducted some of the $CO_2$ measurements, YI provided JMS ocean fluxes, AI provided VISIT fluxes, YN

performed prior flux data processing. NC and PKP prepared the manuscript with contributions from all co-authors. All authors participated in the discussions.

**Competing interests**: The authors declare that they have no conflict of interest.

**Acknowledgments.** We sincerely thank all NOAA and JMA colleagues for providing $CO_2$ measurements used in this inverse modelling study. We sincerely thank Frederic Chevallier, Ingrid Luijkx, Wouter Peters, Christian Rodenbeck at GCP inverse modelling leads for various discussions since 2018, which may have influenced some conclusions derived in this study. This



research has been supported by the Arctic Challenge for Sustainability phase II (ArCS-II; JPMXD1420318865) Projects of the Ministry of Education, Culture, Sports, Science and Technology (MEXT), and Environment Research and Technology
Development Fund (JPMEERF21S20800) of the Environmental Restoration and Conservation Agency of Japan.

**Financial Support.** This research has been supported by the Arctic Challenge for Sustainability phase II (ArCS-II; JPMXD1420318865) Projects of the Ministry of Education, Culture, Sports, Science and Technology (MEXT), and Environment Research and Technology Development Fund (JPMEERF21S20800) of the Environmental Restoration and
Conservation Agency of Japan.

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




## Tables and Figures

**Table 1:** Transport model set up and a priori $CO_2$ emissions used for simulating the atmospheric $CO_2$ concentrations. The range in prior fluxes (2001 – 2020) are also given in PgC yr$^{-1}$ for the study period.

| Model properties | | |
|---|---|---|
| Transport model | MIROC4-ACTM | |
| Resolutions | 2.8 x 2.8º lat-lon grid, 67 vertical layers (surface – 0.0128 hPa) | |
| Meteorology | Nudged with JRA-55 | |
| **Prior Emissions/sinks (2001-2020)** | | |
| **Source/sinks types** | **Abbreviated name** | **Global total: (PgC yr$^{-1}$)** |
| Fossil Fuel and cement production | GridFED | 7.10 – 9.54 |
| Terrestrial biosphere (off-line diurnal cycle by Y. Niwa, using JRA-55 meteorology) | CASA-3hr | 0 (Net annual flux) |
| | VISIT-3hr | -5.9 – (-6.2) |
| Fires | GFEDv4s | 1.7 – 2.2 |
| Ocean exchange | TT09 | -1.41 (Cyclostationary) |
| | JMA | -1.55 – (-2.78) |





**Table 2.** List of 16 inversions based on the combinations of four different prior flux uncertainty (PFU) and measurement data uncertainty (MDU) cases. Every PFU and MDU cases are run for the two prior flux distributions, 1) gc3t (GridFED + CASA-3hr + TT09) and 2) gvjf (GridFED + VISIT-3hr + JMA ocean + Fires - GFED). The abbreviation of inversion cases is arranged as PFU_MDU_Prior Flux.

| PFU (PgC/yr) | MDU (ppm) | Inversion cases | |
|---|---|---|---|
| | | gc3t | gvjf |
| ctl land = 2 ; ocean = 1 | ux2 = 0.1 + 2 × √RSD or ux4 = 0.1 + 4 × √RSD RSD: Residual standard deviation | ctl_ux2_gc3t ctl_ux4_gc3t | ctl_ux2_gvjf ctl_ux4_gvjf |
| gpp_v2 land=0.2-2.0 ; ocean=0.50 | | gpp_v2_ux2_gc3t gpp_v2_ux4_gc3t | gpp_v2_ux2_gvjf gpp_v2_ux4_gvjf |
| gpp_v3 land=0.4-3.0 ; ocean=0.75 | | gpp_v3_ux2_gc3t gpp_v3_ux4_gc3t | gpp_v3_ux2_gvjf gpp_v3_ux4_gvjf |
| gpp_v4 land=0.6-4.0 ; ocean=1.0 | | gpp_v4_ux2_gc3t gpp_v4_ux4_gc3t | gpp_v4_ux2_gvjf gpp_v4_ux4_gvjf |

**Table 3**: Comparison of global total land and ocean $CO_2$ exchanges estimated by inversion model with those from the IPCC 2021 (Table 5.1; (Canadell et al., 2021). The inversion uncertainties due to PFU and MDU for each inversion family based on priors represent ±1σ decadal estimates from the individual inversions. The inversion means and 1-sigma standard deviations are shown for two prior flux cases (8 each for gc3t and gvjf) and all 16 ensemble members (ensm). The IPCC global land and ocean $CO_2$ uptakes are corrected for the river export fluxes. This correction to the net ocean and land uptakes is required because the inverse model estimates fluxes across the land/ocean-atmosphere interface, while the part of the land carbon exchange is exported to the ocean via the rivers and streams.

| $CO_2$ budget | 2000–2009 (PgC yr$^{-1}$) | 2010–2019 (PgC yr$^{-1}$) |
|---|---|---|

|  | IPCC (2021) | MIROC4-ACTM | IPCC (2021) | MIROC4-ACTM |
|---|---|---|---|---|
| FFC Emissions | 7.7 ± 0.4 | 7.99 | 9.4 ± 0.5 | 9.67 |
| Atmospheric increase | 4.1 ± 0.02 | 4.33 ± 0.08 (gc3t) <br> 4.10 ± 0.06 (gvjf) <br> 4.22 ± 0.14 (ensm) | 5.1 ± 0.02 | 5.28 ± 0.08 (gc3t) <br> 5.02 ± 0.07 (gvjf) <br> 5.15 ± 0.15 (ensm) |
| Ocean uptake <br> River export <br> Ocean exchange | -2.1 ± 0.5 <br> + 0.8 <br> = -1.3 ± 0.5 | -1.52 ± 0.03 (gc3t) <br> -1.40 ± 0.11 (gvjf) <br> -1.46 ± 0.10 (ensm) | -2.5 ± 0.6 <br> + 0.8 <br> = -1.7 ± 0.6 | -1.66 ± 0.05 (gc3t) <br> -1.58 ± 0.23 (gvjf) <br> -1.62 ± 0.17 (ensm) |
| Land uptake + <br> Land-use change <br> (= net flux) <br> River export <br> Land exchange | -2.9 ± 0.8 + <br> 1.4 ± 0.7 <br> (= -1.5 ± 1.06) <br> - 0.8 <br> -2.3 ± 1.06 | -2.14 ± 0.1 (gc3t) <br> -2.49 ± 0.14 (gvjf) <br> -2.31 ± 0.21 (ensm) | -3.4 ± 0.9 + <br> 1.6 ± 0.7 <br> (= -1.8 ± 1.14) <br> - 0.8 <br> -2.6 ± 1.14 | -2.73 ± 0.11 (gc3t) <br> -3.07 ± 0.26 (gvjf) <br> -2.90 ± 0.26 (ensm) |

**Table 4.** Mean ± 1-σ standard deviation of annual net land-atmospheric exchange of $CO_2$ (in PgC yr$^{-1}$) from predicted fluxes
for 15 land regions by decade. The predicted fluxes are shown for the best estimates, obtained from the ensemble mean of all
16 inversion cases based on different priors (gc3t and gvjf) and uncertainties (MDU and PFU). The ±1-sigma in the decadal
mean fluxes denotes the range of uncertainty. The estimations for 2000s are compared with the REgional Carbon Cycle
Assessment and Processes phase 1 (RECCAP-1).

| Regions | 2000s (2000-2009) | Fluxes from RECCAP (2000-2009) | 2010s (2010-2019) |
|---|---|---|---|
| Boreal N. America | -0.36 ± 0.09 |  | -0.39± 0.11 |
| Temp. N. America | -0.61 ± 0.13 | -0.47 ± 0.28 (King et al., 2015) | -0.58 ± 0.14 |
| Tropical America | -0.11 ± 0.11 |  | -0.16 ± 0.11 |
| Brazil | 0.04 ± 0.10 | 0.13 ± 0.29 | -0.01 ± 0.07 |





| Temp. S. America | -0.07 ± 0.13 | (Gloor et al., 2012) | -0.12 ± 0.16 |
|---|---|---|---|
| Europe | -0.08 ± 0.11 | -0.20 ± 0.07 (Luyssaert et al., 2012) | 0.07 ± 0.12 |
| Northern Africa | -0.03 ± 0.08 | -0.61 ± 0.58 (Valentini et al., 2014) | -0.08± 0.06 |
| Central Africa | 0.03 ± 0.18 | | -0.10 ± 0.13 |
| Southern Africa | 0.02 ± 0.13 | | 0.07 ± 0.12 |
| Russia | -0.31 ± 0.04 | -0.55 – -0.66 (Dolman et al., 2012) | -0.39 ± 0.06 |
| West Asia | -0.04 ± 0.04 | | -0.08 ± 0.06 |
| South Asia | -0.15 ± 0.09 | -0.15 ± 0.23 (Patra et al., 2013) | -0.2 ± 0.12 |
| East Asia | -0.40 ± 0.07 | -0.41± 0.14 (Piao et al., 2012) | -0.58 ± 0.11 |
| Southeast Asia | -0.10 ± 0.11 | N. A. | -0.15 ± 0.18 |
| Oceania | -0.08 ± 0.06 | 0.04 ± 0.03 (Haverd et al., 2013) | -0.15 ± 0.09 |





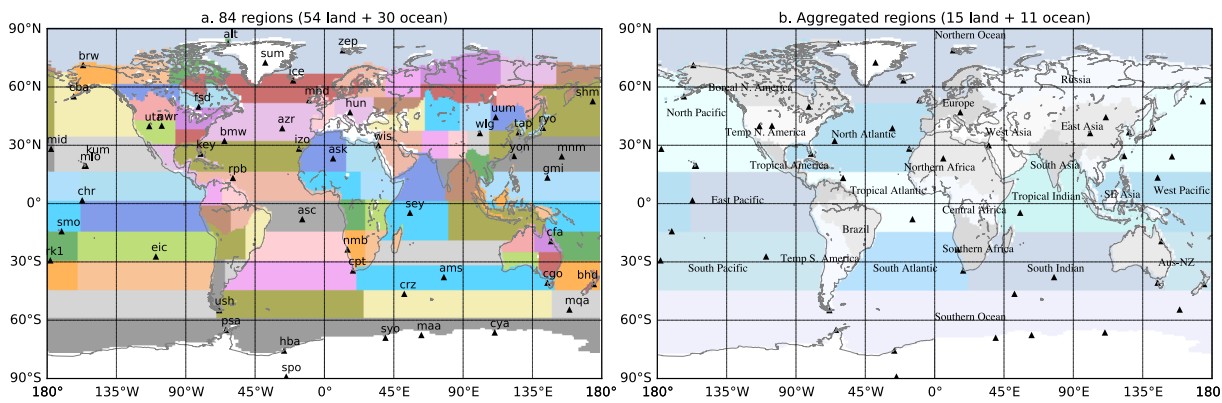

**Figure 1.** Divisions of 54 land and 30 ocean regions for inverse modelling and 50 observation sites (a), the lower panel (b) shows the analysis regions adapted in this study (15 land and 11 ocean), which are consistent with the boundaries of the RECCAP, phase 2 (RECCAP2) land regions (Ciais et al., 2020) and TransCom ocean regions (Gurney et al., 2002).

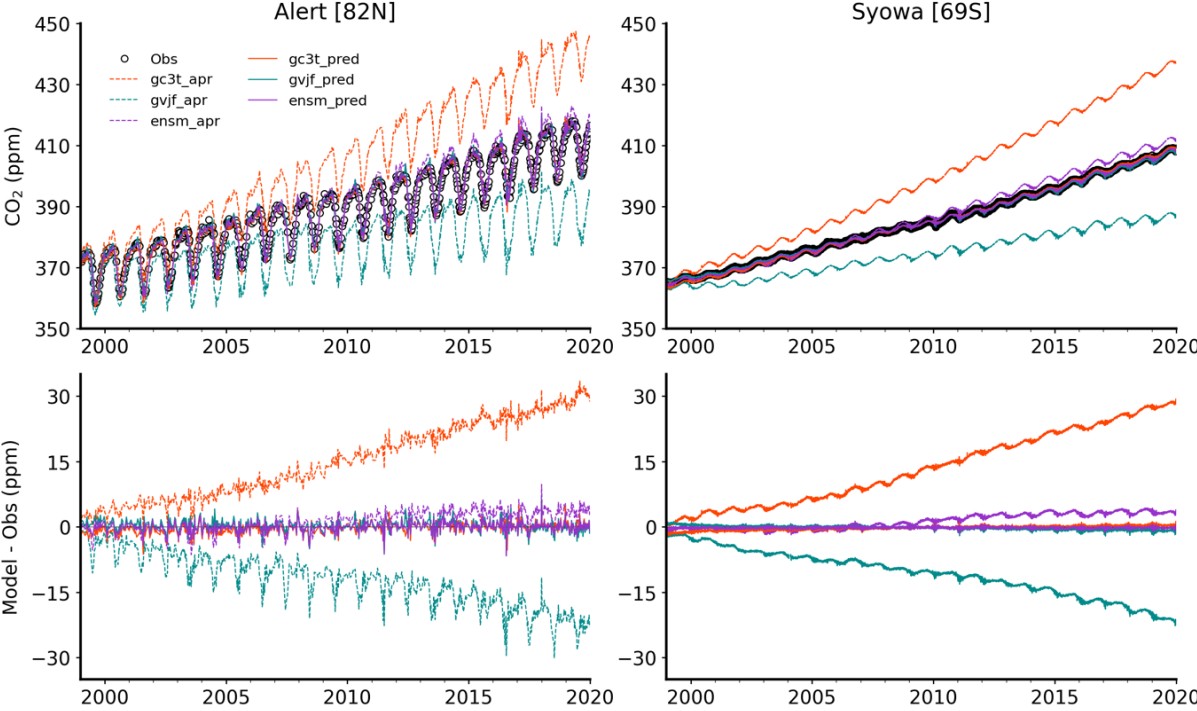

**Figure 2.** Time series of observed and model $CO_2$ time series for two a priori simulation cases (gc3t and gvjf), and inversions using control MDU and PFU case ux4 (top row). Results for the mean of two a priori flux cases and ensemble mean inversion case are also shown. The model-observation differences are shown in the bottom row. Results are shown for two sites, Alert





(82°N, 63°W; measurements by Environment Canada), and Syowa (69°S, 40°E; measurements by Tohoku University/National

Institute of Polar Research).

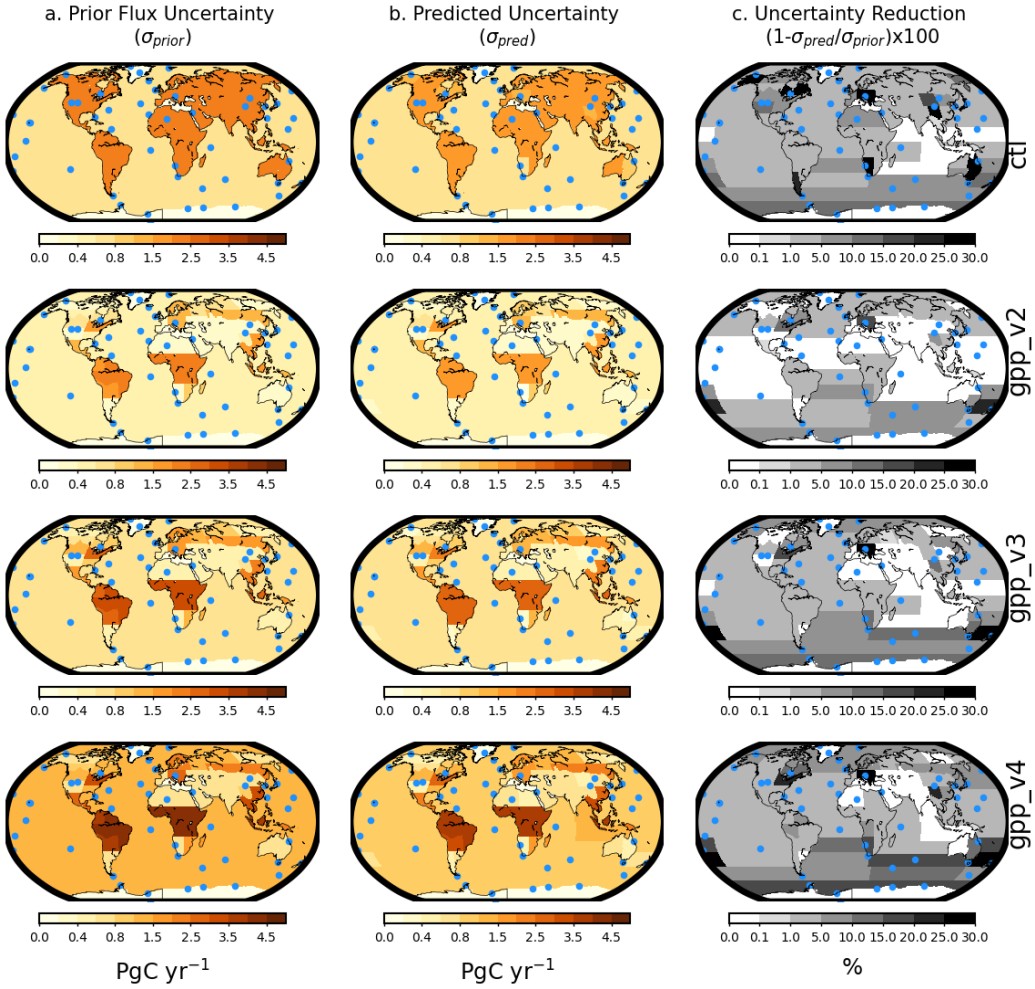

**Figure 3:** Prior and predicted flux uncertainties for four different PFU (4 rows: 'ctl', 'gpp_v2', 'gpp_v3', 'gpp_v4') cases are

shown in the left and middle column, respectively, and the flux uncertainty reduction (FUR) in the right column. The FUR is

an indicator of constraint provided by atmospheric observations. High values (towards 100) in uncertainty reduction indicate

strong data constraints, while low values (close to 0) indicate that the data are not able to move the estimates away from the

prior. All cases are corresponding to the MDU case ux4 and "gc3t" inversion case. The circles show the 50 observation sites

used for inverse analysis.



**Figure 4.** Spatial distributions of $CO_2$ prior flux for two different sets of land and ocean flux combinations (a, d). The middle (b, e) and bottom (c, f) panels show predicted $CO_2$ fluxes by using two different prior flux uncertainty patterns (ctl, gpp_v2).





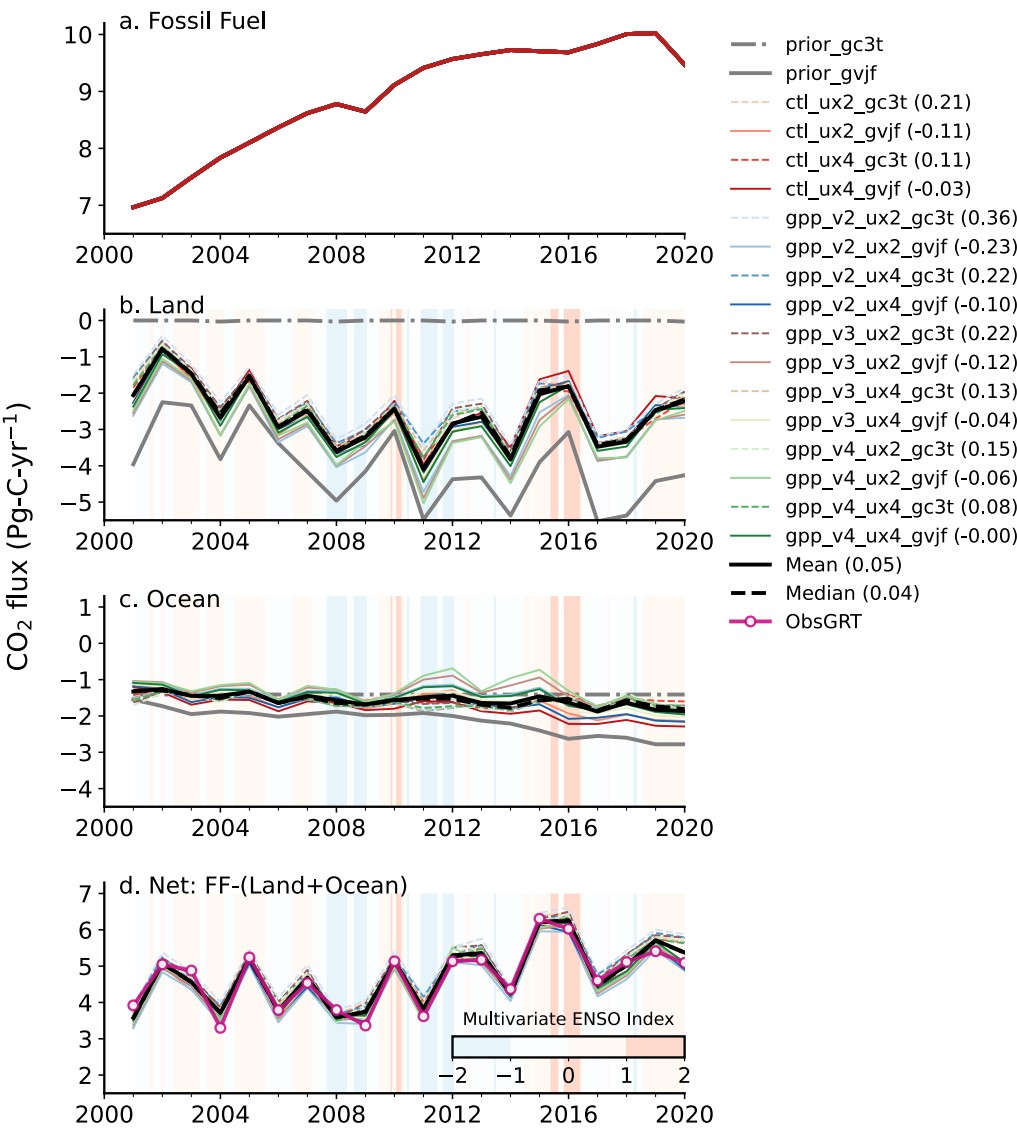

**Figure 5.** Global annual $CO_2$ emissions from fossil-fuel (a) and estimated land (b) and ocean (c) carbon sink from two sets of prior and 16 sets of predicted fluxes based on different combinations of priors (gc3t and gvjf), prior flux uncertainties (PFU: ctl, gpp_v2, gpp_v3, gpp_v4) and measurement data uncertainties (MDU: ux2, ux4). Negative values are uptake by land/ocean, and positive values are sources to the atmosphere. The net flux is calculated by subtracting the total sink (land + ocean) from the fossil fuel and compared with the observed global growth rate of atmospheric $CO_2$ concentration from the NOAA/ESRL (magenta line in panel "d": Dlugokencky and Tans, 2020). The atmospheric $CO_2$ growth rate is converted using a factor 2.13





150    PgC ppm⁻¹. Numbers in the bracket in the legend are budget imbalance between inversions and observed $CO_2$ growth rate. The background shading shows Multivariate El Niño Southern Oscillation (ENSO) Index (Wolter and Timlin, 2011).



**Figure 6.** Interannual variation of terrestrial land $CO_2$ fluxes derived from 16 sets of inversion. The variations are shown for 15 aggregated regions shown in the map. The negative values are counted as $CO_2$ removals from the atmosphere, while $CO_2$ emissions are counted positively.

**Figure 7.** Interannual variation of sea-air $CO_2$ fluxes derived from 16 sets of inversion. The variations are shown for 11 aggregated regions shown in the map. The negative values are counted as $CO_2$ removals from the atmosphere, while $CO_2$ emissions are counted positively.

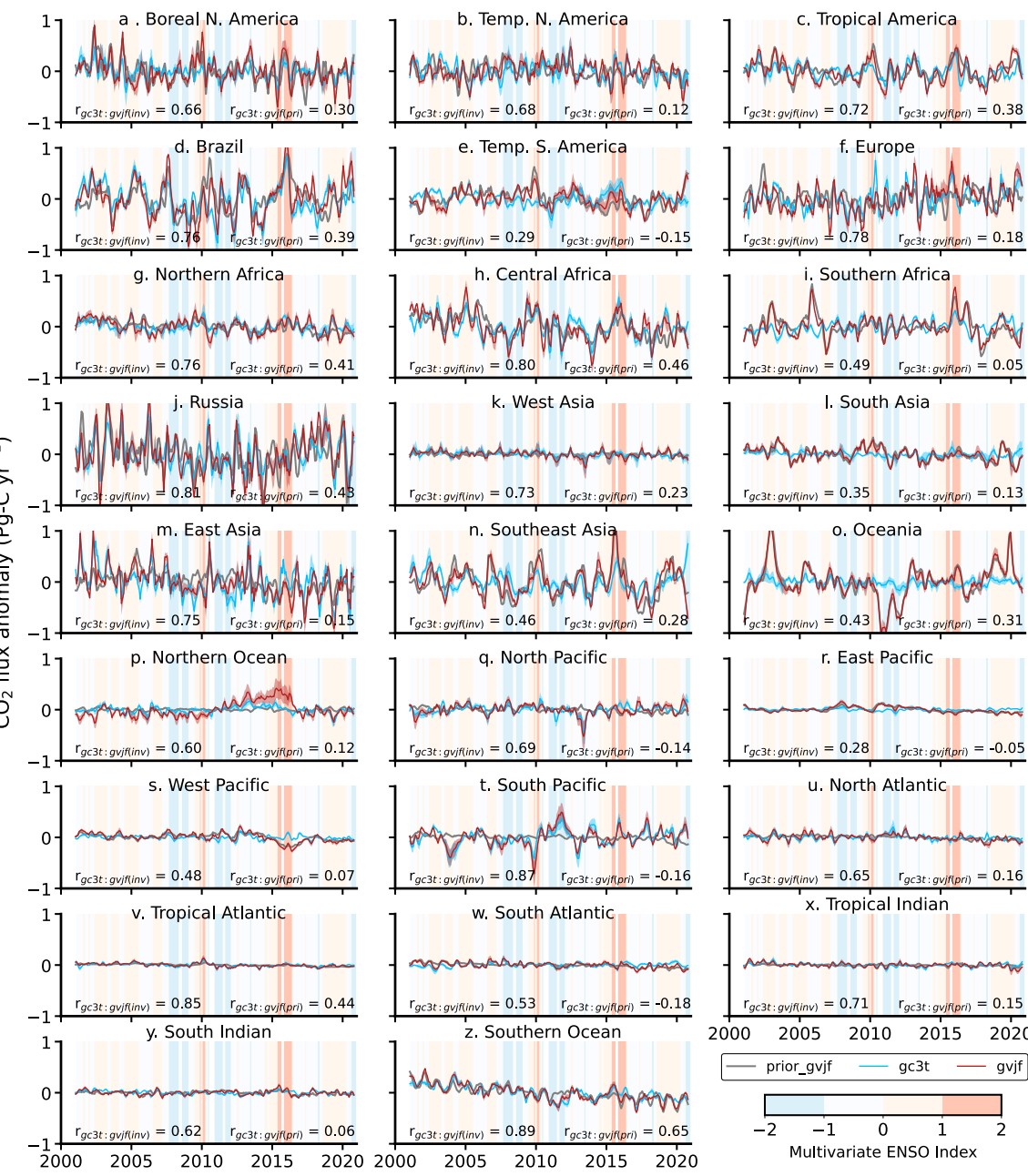

**Figure 8.** Three-monthly running averages of $CO_2$ flux anomalies as estimated by TDI calculation for land and ocean regions from atmospheric $CO_2$ data, with varying prior flux uncertainty and different a priori emissions. Flux anomaly is calculated by subtracting an average seasonal cycle from 1999 to 2020 from the monthly-mean $CO_2$ fluxes. The cases shown in the Figures are obtained by 'gvjf' prior (thick grey line), an average of all 'gc3t' inversion cases, and 'gvjf' inversion cases. The





±1σ around the mean of both inversion cases are shown as shading. The ENSO index is plotted in the background for the corresponding period. The correlations between fluxes (predicted and prior) and ENSO index are shown in Table S3.

1170







**Figure 9.** Seasonal variations in monthly-mean $CO_2$ fluxes at regional scales over 15 lands (upper 4 rows) and 11 oceans (lower 3 rows) regions, along with global land and ocean totals. We prepare the average (2001 - 2020) separately for inversion ensemble cases based on "gc3t" (broken line) and "gvjf" (solid line) prior. The shade around means shows $\pm 1\sigma$ STDEVs for the respective inversion cases. The individual inversion cases are plotted in supplementary (Fig. S2). Note that all panels use different y-scale.

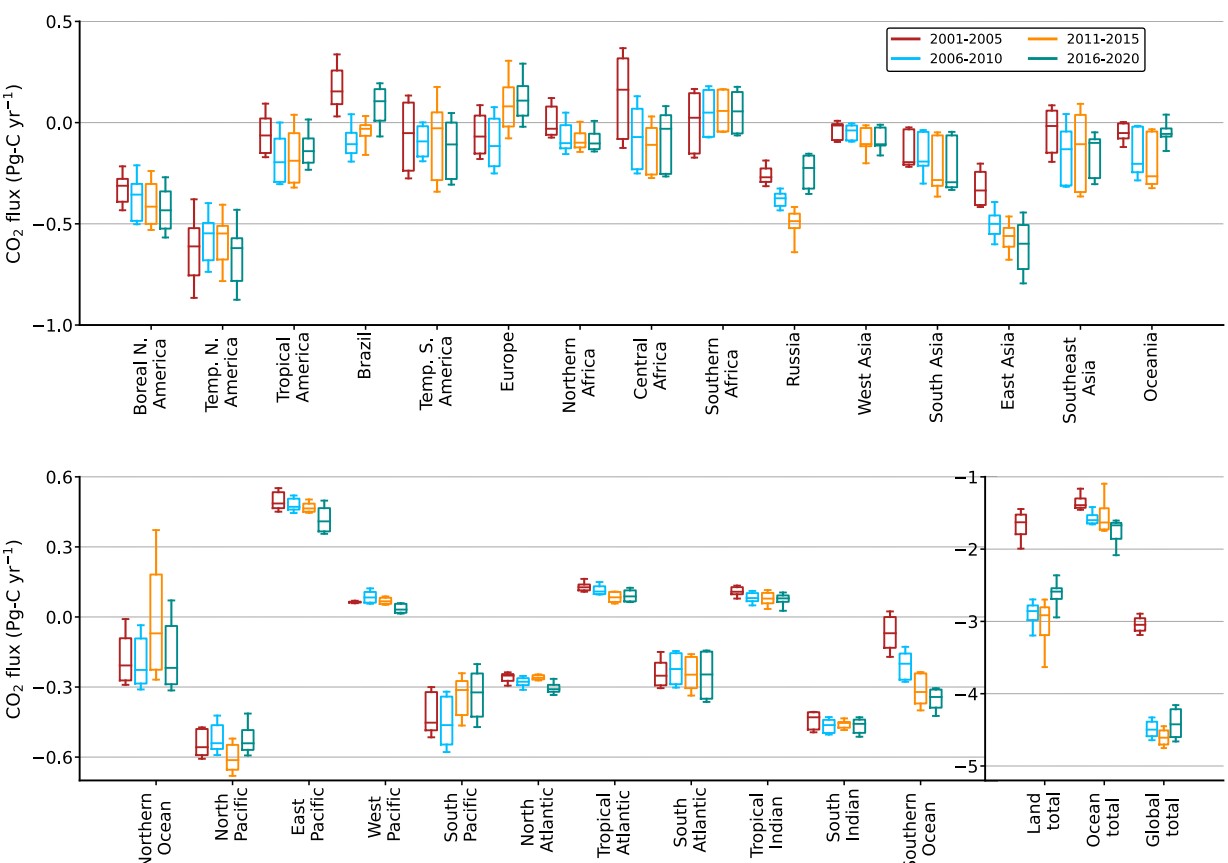

**Figure 10.** $CO_2$ fluxes for land and ocean sink using 16 set of inversion cases. Box plots represent inversions for different time periods. Each inversion cases, based on different prior, PFU, and MDU, are first averaged for selected time period. The box shows 25 and 75 percentiles spread, and the vertical line shows 5 and 95 percentiles from the mean $CO_2$ flux of 16 ensemble inversion cases. Horizontal lines inside box plots denote the median $CO_2$ flux of 16 inversion ensembles. Mean and spread for the whole study period (2001-2020) are given in Table S2.



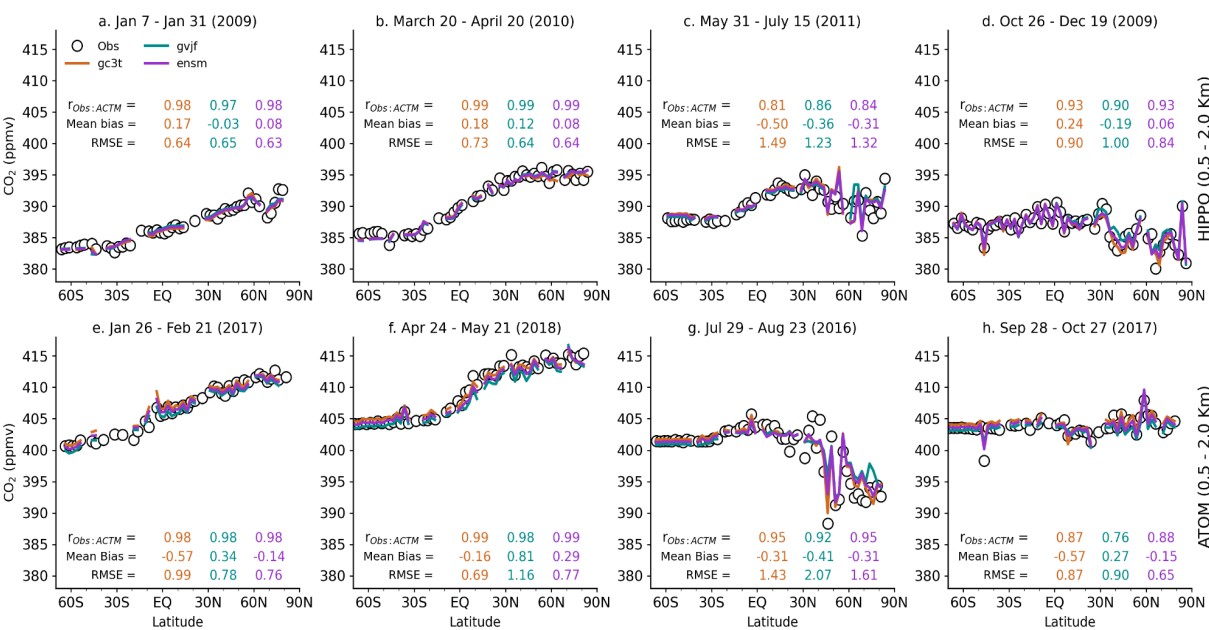

**Figure 11.** Observed and modelled meriodnal $CO_2$ distribution during HIPPO and ATom campaigns. We aggregate the observed and simulated mole fractions for the lower troposphere (between 0.5 and 2.0 Km) at 2.5-degree latitudinal bins. The correlation coefficient, mean bias, and root-mean-square error (RMSE) are also shown in each plot.

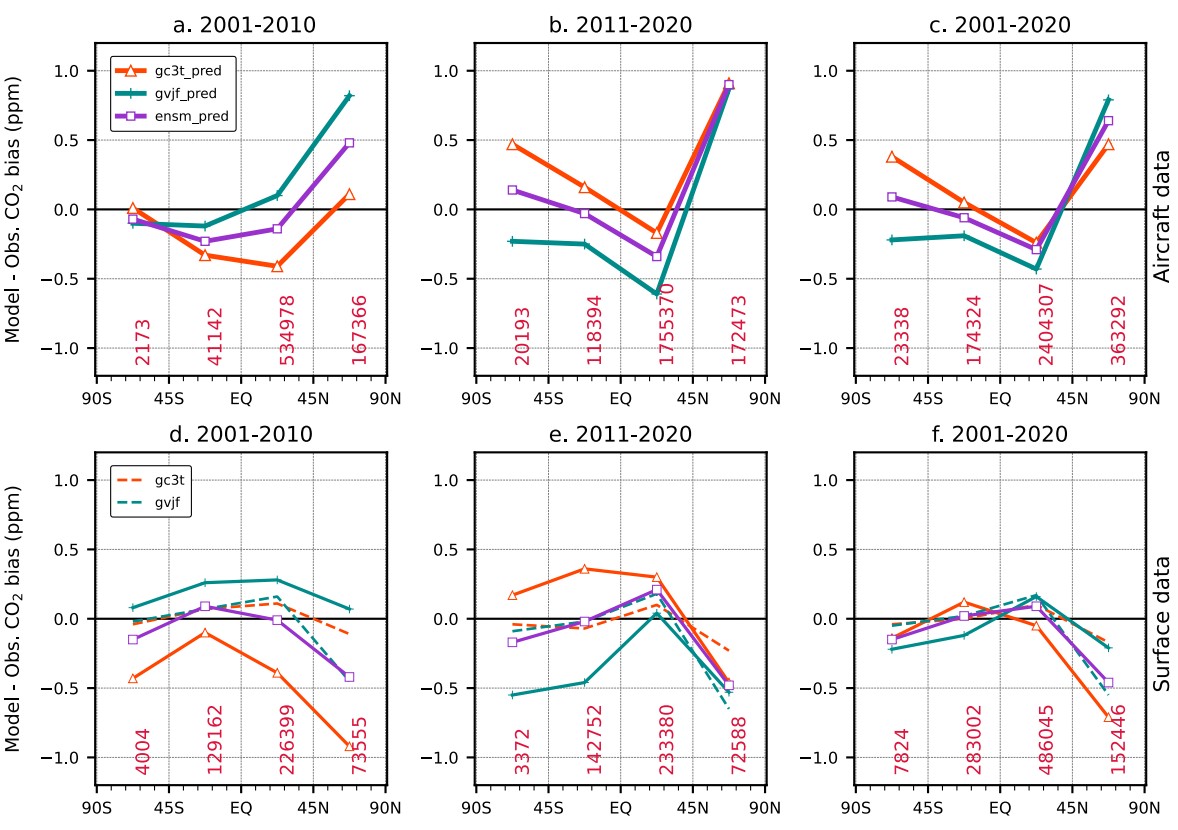

**Figure 12.** Evaluation of the atmospheric inversion products. The mean of the model minus observations is shown for four latitude bands in three periods: **(a,d)** 2001–2010, **(b,e)** 2011–2020, **(c,f)** 2001–2020. The simulations from three inversion cases are compared to independent $CO_2$ measurements made onboard aircraft (a-c) over various parts of the world between 2 and 8 km above sea level and 50 surface sites (b-e). Aircraft measurements archived in the Cooperative Global Atmospheric Data Integration Project (CGADIP, 2020) from sites, campaigns, or programs covering at least nine months between 2001 and 2020, have been used to compute the biases of the differences in four 45° latitude bins. Land and ocean data are used without distinction. The number in each panel shows the total number of data points used for computing bias for each latitude bin.