# Peer review of "Estimated regional CO2 flux and uncertainty based on an ensemble of atmospheric CO2 inversions"

_Atmospheric Chemistry and Physics, 2021_

## Author Comment (AC1)

This manuscript presents inverse model estimates of global and regional CO2 fluxes over the last two decades. The inverse model is based on a single transport model assimilating observations from 50 sites. A series of 16 model simulations is conducted by varying the prior fluxes and prior and observational errors. Results are evaluated against independent aircraft data. The authors found that the ensemble mean of 16 optimized fluxes outperformed individual model outputs. The spread of flux estimates from these 16 model simulations is considered as the uncertainty of the estimated fluxes.

We sincerely thank the reviewer for carefully reading our manuscript and providing us important feedbacks. We have tried our best to address them. Please find answers below each comment. Please find our detailed replies in black to each comment in grey.

General comments
The manuscript presents a detailed study focusing on the inverse model estimation (using a single model) of CO2 fluxes on a global scale for two decades. Therefore, the paper is worthy of publication in ACP after addressing the concerns listed below.

Thank you very much for appreciating our study.

Authors should present the novel aspect of this manuscript. This study uses a single inverse model and conducts a series of model simulations by changing model components, keeping the same observational dataset. Many model intercomparison projects (TransCom and GOSAT and OCO-2 inverse model intercomparisons) address the same aspects by including different transport models but by changing individual model components. Calculating the ensemble mean and spread using a single transport model is not the right way of quantifying the mean and uncertainty in CO2 flux estimates (by not accounting for transport errors).

The novelty of this work is to understand the impact of prior fluxes, and uncertainties (model data uncertainty and prior flux uncertainty) on the estimate of posterior fluxes at the global and regional scale. We agree that the single model transport is not ideal, but please note that this study do not aims to give a full flux estimation uncertainty range, including that due to model transport. We aim to estimate the best $CO_2$ flux with our transport and provide uncertainty on the best-estimated flux. The transport errors are

already explored extensively in the TransCom, GOSAT, and OCO-2 model intercomparison projects. We evaluate fluxes using independent flux estimation such as RECCAP and compare the simulation of a posteriori flues with independent aircraft data like GCP. These are mentioned in the Abstract.

The MIROC4-ACTM model transport quality has been checked separately using multiple tracer simulations as discussed in the methods section. The performances for inter-hemispheric gradient and vertical transport in the upper troposphere and lower stratosphere are reasonable. Nevertheless, no uncertainty is given to our flux estimation system.

We have now revised the manuscript and do our best to highlight the novelty of the work.

To investigate the impact of different modeling components such as model transport, priors, and specification of uncertainties, there could be other systematic approaches, such as designing a series of simulations and quantitatively assessing the uncertainty components. For example, see Basu et al. (2018) and Philip et al. (2019). More rigorous experiments are required if this manuscript intends to assess the spread from priors and prior/observation uncertainties.

We agree approaches in Basu et al. (2018) and Philip et al. (2019) are good, but for full scale transport error uncertainty estimate. However, the uncertainty we estimate is different from that estimated in Basu et al. (2018) and Philip et al. (2019). Our aim is to estimate uncertainty due to choices of prior fluxes and representation of model data and flux uncertainties. It is impossible to estimate the role of inverse model input parameters without running a single model with a number of different choices of inverse model parameters (e.g., prior flux uncertainty, model data uncertainty etc.); hence we use a well-tested single transport model. As stated earlier we compare our estimated fluxes with regional fluxes from independent studies (RECCAP).

Randomly selecting two different terrestrial biosphere models (TBMs) or ocean models is insufficient. Otherwise, reconsider the focus of the manuscript. This study mainly tests land flux scenarios with and without interannual variability (IAV) (CASA versus VISIT). They should consider using different TBMs as priors (diagnostic/prognostic/with and without IAV etc.) with significant regional differences. That can lead to a reasonable spread in the optimized fluxes. Also, how about conducting a sensitivity test by artificially imposing zero net annual flux in the VISIT model?

While making choices on prior flux selection we did look in to the VISIT land and JMA ocean fluxes in comparison with typical DVGM simulations (GCP's TRENDY) and other ocean observation based flux products used in IPCC AR6, respectively. As you can see from the two plots below our Prior fluxes (CASA and VISIT; Takahashi and JMA) are fairly well encompassing the other available products. Therefore, we did not believe we needed more variety in our prior fluxes.

Figures 4, 5, 6 and 7 in the article are testimony that the inversions are able to bring a posteriori fluxes to a common value from both the prior flux extremes.

[Figure]

Figure 1. Comparisons of VISIT and CASA prior cases with TRENDY DGVMs (Sitch et al., 2015; https://doi.org/10.5194/bg-12-653-2015) for 15 land regions.

[Figure]

Figure 2. Comparisons of JMA and Takahashi prior cases with pCO2 observation-based products (Fay et al., 2021; https://doi.org/10.5194/essd-13-4693-2021) for 11 ocean regions.

The manuscript should be written more carefully, especially the introduction and conclusion sections. There are many empty/loose sentences, no connection between paragraphs, introduction not providing any motivation of the paper (it also discusses unrelated aspects), grammatical mistakes, etc., throughout the manuscript. See some of the corrections in the technical-correction section below.

Thank you for pointing this out here and suggesting numerous corrections below. The reviewer#2 is also very kind in reading the manuscript line by line and pointing us to all corrections that are needed. We have made our best effort to incorporate all suggestions and revise the manuscript.

Specific comments

Line 17-21: These two sentences are not connected. You state that model errors and insufficient observations lead to uncertainties in regional flux estimates. However, it is unclear how you address these with your simulations using a single model. State clearly what uncertainty component you are addressing here in this article.

We have revised the two sentences as

"However, the uncertainties in the regional flux distributions remain unconstrained due to the lack of high-quality measurements, uncertainties in model simulations, and representation of data and flux errors in the inversion systems. Here, we assess the representation of data and flux errors by using a suite of 16 inversion cases derived from a single transport model (MIROC4-ACTM) but different sets of a priori (bottom-up) terrestrial biosphere and oceanic fluxes, as well as prior flux and observational data uncertainties (50 sites) to estimate $CO_2$ fluxes for 84 regions over the period 2000-2020."

Line 26-28: This is just a general statement. Need more clarity here: "Interannual variability and seasonal cycle in CO2 fluxes are more consistently derived for different prior fluxes when a greater degree of freedom is given to the inversion System".

We have slightly revised the sentence as

"Interannual variability and seasonal cycle in $CO_2$ fluxes are more consistently derived for two distinct prior fluxes when a greater degree of freedom (increased prior flux uncertainty) is given to the inversion system."

Line 28-29: In line 261, you mention that fluxes are evaluated with aircraft observations. Are you using surface data as well? "…evaluated the inversion fluxes using independent aircraft and surface measurements not used in the inversion".

Yes, surface data are also used, say in Fig. 12 (bottom row)

Line 28-29: Good if you can make it more quantitative, i.e., add some summary statistics or so: "which raises our confidence in the ensemble mean flux rather than an individual inversion"

We consider this remark and revised the sentence as

"We have further evaluated the inversion fluxes using meridional $CO_2$ distributions from independent (not used in the inversions) aircraft and surface measurements, suggesting that the ensemble mean flux (model-observation mean$\pm1\sigma$ standard deviation = 0.3$\pm$3 ppm) best suited for global and regional $CO_2$ flux budgets than an individual inversion (model-observation 1$\sigma$ standard deviation = 0.35$\pm$3.3 ppm)."

Line 31: It seems like an empty/loose sentence: "Differences between 5-year mean fluxes show promises and capability to track flux changes under ongoing and future CO2 emission mitigation policies."

We have slightly revised the sentence for clarity as
"Using the ensemble mean fluxes and uncertainties for 15 land and 11 ocean regions at 5-year intervals, we show promises and capability to track flux changes toward supporting the ongoing and future $CO_2$ emission mitigation policies."

Line 36-38: Cite IPCC report.

Canadell et al. is the referring to the Chapter of IPCC AR6 which has assessed the TCRE etc. Cited again in the next sentence, instead of this general sentence.

Line 44: Be very clear (solutions to …?): "The sinks on the land and ocean constitute a major component of nature-based solutions".

We modified the sentence as
"The sinks on the land and ocean constitute a major component of nature-based solutions to mitigate the rise in $CO_2$ concentration, as discussed in the IPCC AR6 (Canadell et al., 2021)"

Line 45-46: Cite proper references to support the statement.

The paragraph is rearranged and shifted to Line#57. We have added a reference to Kondo et al. (2020) for this sentence.

Line 45-53: In this paragraph, mention global flux uncertainty first, and then note the regional issues, with some additional details. That is, lines 45-46 should come after line 53.

Rearrangements of the text is done as per your suggestions.

Line 55-69: It is not clear why you need this paragraph. "However, the impacts of biases in FFC emissions on inversion estimated CO2 fluxes remained relatively unexplored". Are you exploring this aspect in this paper? Moreover, this paragraph is written poorly.

The aim of this introduction on FFC aliasing effect to give the background of the discussions related to the flux trends for East Asia; Fig. 6 and Fig. 10. Some revisions are now made based on reviewer#2's suggestions and we have added some text in the previous paragraph to link the topics. Hope the paragraph reads better now.

Line 70-73: I don't quite understand this statement! Who provides the metric, what is that metric? What's the meaning of "metric for evaluation of regional fluxes should be evaluated"? Clarify.

We have now revised this paragraph. The 2$^{nd}$ sentence is deleted in the process. Hope the paragraph read well now. The revised paragraph is

"The GCP annual updates of inversions provide a metric for evaluating inversions using independent measurements, mainly from the aircraft campaigns (e.g., Friedlingstein et al., 2020). Evaluation of predicted fluxes from model-data differences may not be straightforward due to the underlying assumptions of a flux inversion system, e.g., for flux correlation lengths or the radius of influence for the measurements, observational data uncertainty, prior flux uncertainty (Baker et al., 2010; Chevallier et al., 2007; van der Laan-Luijkx et al., 2017; Miyazaki et al., 2011; Niwa et al., 2017; Rodenbeck et al., 2003), while the data assimilation system will fit the model concentrations to the observed values. For example, a model-observation difference within ±1 ppm and/or vertical concentration gradient simulation within 1-σ standard deviation of the observed gradient resulted in more than 1 PgC yr$^{-1}$ flux differences between models at regional or sub-hemispheric scales (Gaubert et al., 2019; Stephens et al., 2007; Thompson et al., 2016). Another way of improving our knowledge about uncertainties in regional flux estimations is to employ multiple types of datasets from both bottom-up and top-down modelling systems (Ciais et al., 2021; Kondo et al., 2020), which we have adapted here for checking the regional inversion fluxes, in addition to the GCP like evaluation using independent aircraft data."

Line 71-73: Is this something new? "…should be evaluated using a new transport model simulation of the predicted fluxes, not using the assimilated CO2 field". Be clearer with sufficient details. Most evaluations in current published works are based on model simulation of optimized fluxes. For evaluation, using a

different transport model than the one used in the inversion (as a forward model) is advantageous (not sure if this is what you mean here). Also, are you exploring this in this manuscript/study?

No, unfortunately this could not be explored in details and remained as a hypothesis. Thus, we have revised this paragraph as mentioned above by cleaning up texts.

Line 73-81: I'm lost here. From re-reading this, I understand that the assessment of the spread of optimized fluxes obtained by conducting multiple simulations using different model inputs is a better way of quantifying the uncertainty than simply evaluating the optimized $CO_2$ concentrations against independent measurement data. Revise the entire paragraph to be more apparent.

Thank you. We have revised the paragraph as a whole, and added at the end as discussed in previous comment (Line 70 – 73)
"Another way of improving our knowledge about uncertainties in regional flux estimations is to employ multiple types of datasets from both bottom-up and top-down modelling systems  (Ciais et al., 2021; Kondo et al., 2020), which we have adapted here for checking the regional inversion fluxes, in addition to the GCP like evaluation using independent aircraft data."

Line 82-85: These uncertainty sources have been investigated previously. Cite some of those critical studies here.

We have cited : (Basu et al., 2018; Patra et al., 2005a; Philip et al., 2019; Qu et al., 2021; Wang et al., 2018).

Line 92-95: This statement is not correct: "Such intercomparisons used single inversion from different modeling groups and provided the range in $CO_2$ flux uncertainty due to differences in transport models.". These intercomparisons assessed uncertainty arising from different model components, not just the transport model differences. For example, see Crowell et al., 2019 and Peiro et al., 2022.

We are aware of these publications (Crowell et al. 2019 is already cited; Peiro et al. used fluxes for an extended period) which makes assessments of regional fluxes estimated by inversions using OCO-2 data but none of both give separate assessments of prior flux uncertainty vs data uncertainty on the inversion results, for example. In fact, it is impossible to estimate the role of inverse model input parameters without running a single mode with a number of different choices of inverse model parameters (e.g., prior flux uncertainty, measurement data uncertainty etc.).

Following your and Reviewer#2 suggestion we have modified the statement as

"Such intercomparisons used single inversions from different modelling groups and provided the range in total $CO_2$ flux uncertainty due to the choices of prior fluxes distribution, prior flux uncertainty, observational data uncertainty, and the model transport uncertainties."

Line 123-124: This sentence is not clear to me.

The sentence is revised as

"Reasonably good model transport in MIROC4-ACTM enables us to use any mismatch between observation and simulations to estimate the land and oceanic fluxes using the inverse modelling technique (details in Section 2.4)."

Line 128 and 129: Just "used" not "simulated" (?): "… is simulated using …"

Corrected as per your suggestion.

Line 135: "…downscaled to 3-hourly time intervals…": Mention how you downscaled; which variable used; and cite proper literature.

Thank you for pointing this out. The sentence is revised as "The CASA and VISIT monthly-mean fluxes are downscaled to 3-hourly time intervals by redistributing respiration and gross primary production (Olsen and Randerson, 2004) using JRA-55 meteorology, i.e., 2m air temperature and incoming solar radiation at the earth surface"

Line 136: Double-check if it is version 4.1? "…fire emissions are used from GFEDv4s (van der Werf et al., 2017…".

Thank you for catching the mistake. Yes, we have used v4.1s

Line 145: Complex notations: gc3t and gvjf. What is "3" and "t" in gc3t?

Agreed, but we created a 4 lettered name to accommodate all 4 fluxes, as given in Eq. 1. We have slightly rearranged for better clarity about how the notations are formed: g: GridFED FFC, c3:CASA-3hr, v:VISIT, t:TT09 ocean, j: JMA ocean, and f: Fire

Line 149-150: Revise: "to evaluate the strength of MIROC4-ACTM simulations to derive fluxes consistently". How do you evaluate the strength of simulation? Why did you mention "consistently" here? Fluxes will be derived using the inverse model, so how can you "evaluate the strength of forward simulation"?

Sorry for the unclear information. Our intent was to evaluation inversion strength. So, we have changed "simulations" by "inversions" for clarification. We stated both - consistently (or the lack of it)! We think transport model is key for the fluxes we derive, although $CO_2$ is an inert species and transport is linear. So, we thought of mentioning the model's name.

Line 159: Cite proper references: "WDCGG websites as appropriate"

Given as : GML/NOAA (https://gml.noaa.gov/aftp/data/trace_gases/co2/flask/) and WDCGG (https://gaw.kishou.go.jp/) websites

Line 162: Is this the grid cell with the observation location? "…nearest grid of observation location at hourly intervals…".

Revised as "the grid point nearest to the observation location" for clarity

Line 164: "These temporal data gaps (1-6 months) are filled using the curve fitting method based on the digital filtering technique". Have you conducted simulations without using curve-fitted data? Why was this data filling necessary?

The matrix inversion system requires data every month of the inversion period. We have checked the fitting program works well when data gap is less than 6 months, as the seasonal cycle is derived by using 6 harmonics.

Line 200-210: How about conducting a simulation with "gpp_v4" along with "ocean PFU = 0.5"? Explain the rationale for selecting different prior error scenarios you considered in this study.

We have stated the rationale as: (sorry without conducting the suggested simulation)
Selection of wide range of PFUs, in the range of $0.5 - 1.0$ PgC yr$^{-1}$ the ocean regions and $0.2 - 4.0$ PgC yr$^{-1}$ for the land regions allows us to understand about the stability of the inversion system as assess the range of a posteriori fluxes for aggregated sub-continental/basin regions or the land and ocean totals.

Line 234: "High values (FUR towards 100)": If FUR is in percentage, then revise the equation in line 233.

Revised as per your suggestion.

Line 244: Not clear: "… indicative of the observational constraint regional fluxes…"
Line 245: "…we recommend that the spread of ensemble inversions provide more representative estimation of the regional CO2 sources and sinks.". "Spread" represents "a measure of uncertainty", not a "representative estimation of…". Why do you add "recommend" here?

This sentence is revised as "As discussed later in this article, the FUR is only indicative of the observational constraint on the regional fluxes, the spread of ensemble inversions provides a measure of uncertainty of the regional CO$_2$ sources and sinks."

Line 309 and 311: Revise this sentence: "Hence, the magnitude of biases and RMSE indicates predominantly the accuracy of the predicted fluxes.". Model transport is one of the sources leading to uncertainties in the predicted fluxes.

As a remark. we have revised this sentence as "Hence, the magnitude of biases and RMSE indicates predominantly the accuracy of the predicted fluxes (the errors due to model transport and measurement network are not explored in this study)."

Line 649: "CO2 simulations are derived from three sets of prescribed fluxes: "gc3t", "gvjf", and "ensm".": I'm assuming that the evaluation is conducted for all 16 inversions (?).

All 16 sets are not simulated by model, but we used 3 representative cases, and part of this sentence is now revised for clarity as
"The CO$_2$ simulations are derived from three sets of prescribed fluxes: "gc3t" (case: ctl_ux4_gc3t in Table 2), "gvjf" (case: ctl_ux4_gvjf), and "ensm" (average of all 16 inversions)."

Lines 709-720: I'm not sure if these details (+ Figure S10) are required in this paper.

Thank you for this suggestion. Review#2 also expressed concerns on this part of the discussion.
We have removed this paragraph and the one before it, and Fig. S10.

Line 775-782: Empty/loose sentences.

We have deleted this final paragraph of the Conclusions, as per your suggestion and also because lines 692-720 in the submitted version are deleted following suggestions from you and Reviewer#2.

Technical corrections
Line 14: Better add "atmospheric" here: "chemistry-transport model (ACTM)".
Line 16: Better avoid text in parenthesis: "regional flux (+ve: source to the atmosphere; -ve: sink on land/ocean)".
Line 21: Move the number of the sites (50) from here to the appropriate part of the sentences: "data uncertainties (50 sites)".
Line 24: Is this "22-33% and 16-18%" for land vs ocean? Not sure this is clear enough here.
Line 25: Not clear what this approximate means here: "best estimations for (approx. 2000-2009)".
Line 52: Revise and add more clarity: "partitioning exists greatly in the … release".
Line 55-56: Revise this sentence: "…because inversion calculations do not optimize…".
Line 90-91: You can write these in a better way: "inversions from … for inversions using … or for inversions".
Line 99: Revise: "observed and model data processing".
Line 100: Avoid capital letter: "the Results and discussion".
Line 155-156: Avoid repetition of "from".
Line 1120: Correct this: "lower panel (b)".
Line 242: Correct: "…West Asia, Northern Africa. The Tropical Indian Ocean…".
Line 252: Correct: "as per analysis".
Line 302-307: Use simple notations. For example, avoid "aircraft" from "x".
Line 308: Correct: "CO2 mixing ratios".
Line 317: Use the term "grid cells".
Line 336: Avoid ".": "3.2. Global totals."
Line 346: Use "mean": "Ensemble means land".

We appreciate your help very much. All of the above corrections are made in the revised manuscript.
.

Line 563: Revise: "It is not easy for us to explain".

Thank you for the suggestion, we have further scrutinized the Yasunaka et al. paper and added

"It is not easy to put forward a hypothesis for the weaker sink in summer than in winter of Northern Ocean, while we can speculate that the atmospheric $CO_2$ decrease in polar air exceeds compared to the decrease that occur over the surface sea-water and reduced solubility of $CO_2$ in warmer water. Indeed, Yasunaka et al. (2018) have shown that the Greenland-Norwegian seas and Barents Sea are indeed acts as milder sink of $CO_2$ (flux = -4 to -5 mmol $m^{-2}$ $day^{-1}$) during June-August compared to the October-March (flux = -10 to -15 mmol $m^{-2}$ $day^{-1}$), and the Chukchi Sea and Arctic Ocean show strongest uptake in October. Thus, as whole the Northern Ocean of our study could act as the weakest sink in summer months."

Line 763: Avoid "Please".
Line 766: Correct: "is unanimously located".
Figure 4: Choose a different font that is clearer.
Figure S2: Correct to CO2: "monthly-mean CO fluxes"

All of the above corrections are performed. We appreciate your help very much.

References

Basu, S., et al.: The impact of transport model differences on CO2 surface flux estimates from OCO-2 retrievals of column average CO2, Atmos. Chem. Phys., 18, 7189–7215, https://doi.org/10.5194/acp-18-7189-2018, 2018.

Crowell, S., et al.: The 2015–2016 carbon cycle as seen from OCO-2 and the global in situ network, Atmos. Chem. Phys., 19, 9797–9831, https://doi.org/10.5194/acp-19-9797-2019, 2019.

Peiro, H., et al.: Four years of global carbon cycle observed from the Orbiting Carbon Observatory 2 (OCO-2) version 9 and in situ data and comparison to OCO-2 version 7, Atmos. Chem. Phys., 22, 1097–1130, https://doi.org/10.5194/acp-22-1097-2022, 2022.

Philip, S., et al.: Prior biosphere model impact on global terrestrial CO2 fluxes estimated from OCO-2 retrievals, Atmos. Chem. Phys., 19, 13267–13287, https://doi.org/10.5194/acp-19-13267-2019, 2019.

All of these references are cited in the revised manuscript.

---

## Author Comment (AC2)

This manuscript explores the sensitivity of a global CO2 flux inversion using CO2 mixing ratio measurements to the choices of prior flux, prior flux uncertainty, and measurement uncertainty assumed in the inversion. Gap-filled measurements from 50 globally-distributed sites are used and monthly fluxes across 2000-2020 are estimated for 84 emission regions (54 on land, 30 for the oceans). Given that the fluxes to be estimated are severely under-constrained by the data used here, especially in the tropics and southern hemisphere (SH) were the data are sparse, it is not surprising that the final estimate should depend strongly on the prior estimate assumed going in. The sensitivity to two different sets of prior fluxes are explored here: 1) annually-balanced CASA land biospheric fluxes paired with Takahashi (1999) ocean fluxes, a combination that results in too large of a trend of CO2 in the atmosphere due to the lack of the realistic global land sink, and 2) land biospheric fluxes from the VISIT model that have too large of an global annual uptake, resulting in a too-small trend of CO2 in the atmosphere, coupled with ocean fluxes from the JMA model. The bias in the global land+ocean uptake embodied in each of these sets of prior fluxes is reduced in the posterior flux estimates, but remains at a lower level, especially for individual regions instead of the global level. Since the two priors had errors in the trend of opposite signs, averaging results over the two cases results in lower errors with respect to the truth.

Besides varying the prior fluxes themselves, the authors explore the impact of assuming different values for the uncertainty on these prior fluxes as well as the uncertainty on the measurements (or model-measurement mismatches, to be more precise). One must assume some value for these uncertainties in the inversions, and these assumed values are always incorrect to some degree, since one never knows precisely what the true uncertainty ought to be: the larger the errors in these assumed values, the larger the error in the a posteriori estimate due to the bad assumptions; these errors tend to be systematic rather than random, so it is quite useful to know how large of an impact they have. In my view then, this study is worth publishing because it quantifies the impact of these mis-specified statistical assumptions, even if the global CO2 flux inversion underpinning this work is far from being cutting edge. (Global CO2 inversions of this sort using the in situ CO2 measurement network have been done for over two decades, going back to the 1990s at least. There are now many more in situ measurement sites than the 50 used here, including tall towers on the continents and the routine aircraft profiles that have been used here for evaluation purposes. Furthermore, there are column-integrated CO2 measurements from ground-based Fourier spectrometers looking at the sun, as well as the huge volume of column CO2 data from satellites. These data are now used routinely to estimate fluxes for thousands of regions, instead of just the 84 used here.)

The authors have done a nice job setting up their ensemble of runs (16 total, permutations of the 2 flux priors, 2 different assumptions for the magnitude of measurement uncertainties assumed, and 4 different assumptions for the magnitude of a priori flux uncertainty assumed) and have done a careful job of analyzing the results from a variety of perspectives (global total, land/ocean totals, regional fluxes, annual means, interannual variability, seasonal variability, the estimation uncertainty versus the sensitivity of the estimate to the priors and assumed statistics, and errors evaluated by comparing to independent data). While the manuscript is quite long and may be daunting to some readers, I realize that there is a lot of ground to cover and am sympathetic that the length is not inappropriate. However, my main problem with the manuscript is with the writing: in many places, it is difficult to understand the points that are being made. As a result, I had difficulty understanding precisely what was done in this work, both in terms of the method used for the inversion and the methods used for the analysis, as well as the results obtained and the logic used to interpret those

results. Therefore, before being published in ACP, I would like the authors to do a better job with their writing, making it clearer what was actually done and what the implications of their work really are. I think that they should also note that their setup here is more under-constrained by the data than most, and therefore the impact of the error sources that they examine is probably larger for this study than for inversions that use more data. Finally, when quantifying the uncertainty in the flux estimates, the authors need to do a better job explaining what error terms are quantified by their ensemble spread, and what are not (the authors note that transport model error is not quantified, since they only used a single transport model in this study, but they do not do a good job pointing out the difference between the estimation errors usually quantified by the inversion and the errors examined here in their sensitivity study, or the slight overlap between the two (due to the errors or differences in the prior fluxes)). I have noted below the places where the authors should clarify their text, and I have made numerous editorial corrections and suggestions for better wording that will hopefully make it easier for the reader to understand what is going on. I apologize for not breaking out the more-editorial comments separately from the more substantive ones: at the moment, they are all mixed together in rough line-number order.

We sincerely thank the reviewer for carefully reading our manuscript and providing us important feedbacks. We are overwhelmed by your efforts in reading the article so carefully. We have no words to appreciate or thank you enough. While revising the manuscript and writing replies we have felt that it requires immense patience and extraordinary helping nature to prepare such a review, for no credits.

We have tried our best to address them and revised whole manuscript as per your suggestions. Please find our replies in black below each comment in gray.

Detailed comments (line number indicated):

24: "without riverine export correction" -- I take this to mean that these are the actual fluxes inverted, and that if you corrected for 0.6, say, you would get 1.6 +0.6 = 2.2 PgC/yr storage in the ocean. Please give more detail as to what making this correction would do to the results and how that relates to anthropogenic fluxes/storage.

We have added

"The rivers carry about 0.6 PgC yr$^{-1}$ of land sink in to deep ocean, and thus the effective land and ocean partitioning is -2.3±0.3 and -2.2±0.3, respectively."

29-30: "which raises our confidence in the ensemble mean flux rather than an individual inversion." Reword for clarity.

Revised as

"We have further evaluated the inversion fluxes using meridional $CO_2$ distributions from independent (not used in the inversions) aircraft and surface measurements, suggesting that the ensemble mean flux (model-observation mean±1σ standard deviation = -0.3± 3 ppm) best suited for global and regional $CO_2$ flux budgets than an individual inversion (model-observation 1σ standard deviation = -0.35±3.3 ppm)."

52: what does "greatly" indicate here?  Reword for clarity.

The land and ocean sink uncertainty assessed in Canadell et al. is based on GCP CO2 budget. We revised the sentence as
"The uncertainty in land and ocean sink partitioning of up to about 1 PgC yr$^{-1}$ in the IPCC AR6 are based on the Global Carbon Project (GCP)'s annual carbon budget"

56:  It is not correct to say that inversions do not optimize the FFC emissions.  They solve for corrections to the prior fluxes (including FFC ones), and then this correction must be partitioned between ocean, land biospheric, and FFC fluxes.  Because the uncertainty on the prior FFC fluxes is thought to be much lower than that on the land biospheric fluxes, most of the correction should therefore be attributed to the land biospheric fluxes.  However, a small part of it could also be attributed to the FFC ones.  Usually this small amount is neglected and all of the correction over land is attributed to the land biospheric fluxes.  However, this is a simplification.  Inverse modelers could, without changing their inversions, choose to partition the correction differently between the two.  As it is, they are very aware that some of the correction that they currently attribute to the land biospheric fluxes could also be due, in part, to errors in the initial FFC fluxes.

Following your suggestion, we have revised the later part of the first sentence for better clarity, as
"Top-down inverse models estimate residual natural or non-FFC CO$_2$ fluxes from land and ocean regions because inversion calculations do not explicitly optimise the FFC emissions, i.e., the FFC emissions are not revised, but the a priori land and ocean sinks are revised."

Both of the corrections are made.

We have updated/expanded the discussion here:
Evaluation of predicted fluxes from model-data differences may not be straightforward due to the underlying assumptions of a flux inversion system, e.g., for flux correlation lengths or the radius of influence for the measurements, observational data uncertainty, prior flux uncertainty (Baker et al., 2010; Chevallier et al., 2007; van der Laan-Luijkx et al., 2017; Miyazaki et al., 2011; Niwa et al., 2017; Rodenbeck et al., 2003), while the data assimilation system will fit the model concentrations to the observed values. Thus, good statistics for the validation metric using independent data and assimilated concentration field did not ensure good agreement between the estimated fluxes by different models, at the sub-hemispheric and sub-continental scales, or separately for land and ocean. For example, a model-observation difference within ±1 ppm and/or vertical concentration gradient simulation within 1-σ standard deviation of the observed

gradient resulted in more than 1 PgC yr$^{-1}$ flux differences between models at regional or sub-hemispheric scales (Gaubert et al., 2019; Stephens et al., 2007; Thompson et al., 2016).

84: add "to" after "leading";  add "and" after "error,"
92: change to "single inversions"

Both the corrections are made.

93-95: What you are trying to say here is that none of these studies partition the inversion-group-based uncertainty between these three sources, but just give the total uncertainty.  Try to reword it to bring out that point better.

Following your suggestion, two sentences are merged to one as
"Such intercomparisons used single inversions from different modelling groups and provided the range in total CO$_2$ flux uncertainty due to the choices of prior fluxes distribution, prior flux uncertainty, observational data uncertainty, and the model transport uncertainties."

100: change to "Section 2" and "Discussion"
104: change to "Section 4"
112: remove "(" before "Bisht"
134: change "via" to "due to", for clarity;  correct "on the net a large land sink" -- doesn't make sense now
144: add "fluxes" after "land"
Table 1, line 3: add a degree sign after the first "2.8"
155: change "The 38" to "Of these, 38"
156: "and 3"
162: reword to "sampled at the observation time and the grid box nearest to the observation location at hourly intervals."
163: change "six months" to "six-month"

Thank you for these suggestions. All of the above corrections are made in the revised manuscript.

166: "with six harmonics by a cut-off length of 24 months for the digital filter."
It is not really clear how these six harmonics were chosen, given this wording. Please reword it to be clearer.

Sorry for the incorrect formulation, revised as
 "We fit the measured and simulated time-series at daily-weekly time intervals with six harmonics (extracts the sinusoidal component, i.e., seasonal cycle) and Butterworth digital filter with a cut-off length of 24 months (determines the long-term trends)"

169, Section 2.4:  It is unclear what sort of Transcom-like inversion is being performed here.  Is it the so-called "cyclo-stationary" inversion, in which a single, typical seasonal cycle of flux is being solved for, then added onto the prior?  Or is it a fully time-dependent inversion in which the seasonal cycle for each year is optimized?  How many terms are in

the state vector solved for? Is it a matrix-based inversion? How large is the matrix actually inverted? How is the prior treated in this framework (i.e. what is the set of equations that is actually solved, and where does the prior fit into that)? I note below that equations (1)-(3) do not seem to be written correctly, in that S and D ought to be vectors, not matrices. In Figure S1 it is suggested that the basis functions in the G matrix have only been run out for four months -- how is the impact of a flux represented for times after those four months? Is the influence just ignored? Perhaps I am missing something here -- please describe what you are doing more completely to make all this clearer.

Apologies for the poor construction of the equations and description. It is now revised as:

"In the Bayesian inversion, when the relation between model parameters and data parameters is linear ($d = \mathbf{J}\vec{s}$), the misfit function ($\chi^2$) is constructed as (Rayner et al., 2008; Tarantola, 2005)

$$\chi^2 = \frac{1}{2}\left[(\vec{s} - \vec{s}_0)^T \mathbf{C}(\vec{s}_0)^{-1}(\vec{s} - \vec{s}_0) + \left(\mathbf{J}\vec{s}_0 - \vec{d}_{obs}\right)^T \mathbf{C}(\vec{d})^{-1}\left(\mathbf{J}\vec{s}_0 - \vec{d}_{obs}\right)\right] \qquad (2)$$

Assuming that the elements of $\mathbf{C}(\vec{d})$ are uncorrelated, the solutions for $\vec{s}$ and $\mathbf{C}(\vec{s})$ can then be written as

$$<\vec{s}> = \vec{s}_0 + \left(\mathbf{J}^T \mathbf{C}(\vec{d})^{-1}\mathbf{J} + \mathbf{C}(\vec{s}_0)^{-1}\right)^{-1} \mathbf{J}^T \mathbf{C}(\vec{d})^{-1}(\vec{d}_{obs} - \vec{d}_{ACTM}) \qquad (3)$$

and posterior error covariance

$$\mathbf{C}(\vec{s}) = \left(\mathbf{J}^T \mathbf{C}(\vec{d})^{-1}\mathbf{J} + \mathbf{C}(\vec{s}_0)^{-1}\right)^{-1}$$

where $\vec{s}_0$ is the prior source for the 84 regions and 288 months in 1998-2021, $\mathbf{C}(\vec{s}_0)$ is the prior source error covariance matrix, $\vec{d}_{obs}$ is the measurement data at 50 sites for 288 months, and $\mathbf{C}(\vec{d})$ is the data error covariance matrix. $\vec{d}_{ACTM}$ ($\approx \mathbf{J}\vec{s}_0$) is forward model simulation time series using a priori fluxes, run continuously for the whole period of analysis, and sampled at the time and locations of the individual measurement before calculating monthly means. $\mathbf{J}$ is the Jacobian matrix of sensitivities of observations with respect to $\vec{s}$, calculated using simulations of unitary pulse sources for a month for the 84 basis regions, and sampled at the 50 measurement sites. The unitary pulses are simulated for 4 years and originated for each month of year 2011 for all regions (84 regions × 12 months = 1008 tracers per year; one set of $\mathbf{J}$-matrix is reused for all years). We have shown in Fig. S1 and associated text that use of annually repeating $\mathbf{J}$ does not affect the inversion results significantly as majority of the spatial and temporal flux variabilities are coming from the a priori, which are simulated using interannually varying meteorology. The elements in $\vec{s}$ are the optimised $CO_2$ fluxes (referred to as a posteriori or predicted flux) from 84 regions at monthly time intervals. The off-diagonal elements of $\mathbf{C}(\vec{s}_0)$ are kept zero, assuming the a priori fluxes are uncorrelated to one another regions or time. The correction fluxes ($\vec{s} - \vec{s}_0$ in Eq. 3) is primarily determined by the term ($\vec{d}_{ACTM} - \vec{d}_{obs}$), scaled by the data/flux uncertainty."

173: change "lands" to "land"
178: usually, you would give the cost function a symbol, like:   "J = (D-Gs)T ... etc."

The equations are modified for cost function like symbols.

Note on equations: These need to be cleaned up a bit to conform with standard notation. Vectors should be lower case and bold. Matrices should be upper case and bold. Change this both in the equations and text. At the moment, you

have the fluxes being put into a 2-degree matrix, S, whereas they are usually put into a 1-degree vector, s. Why do you have it as a matrix? Are you putting the vectors for multiple inversion cases all together into one big matrix and doing the inversion all together at the same time across all cases? (If so, the equations given are not correct.) If not, the fluxes should be put in vectors s.

We follow the equations from Rayner et al. (2008) and Tarantola (2005). They are now written in the notations you suggest. The vectors and matrix are shown in small letters with arrow on top and capital letters, respectively.

187-188: A word about how you order the monthly fluxes into vector s (not matrix S) would be useful: the 84 measurements for month 1, followed by the 84 for month 2, etc...?

The inversion code is made available on github, which was first developed by Peter Rayner, Rachel Law et al. at CSIRO, and later distributed through TransCom inversion activities by Kevin Gurney, Rachel Law et al. We have revised some of the codes and functionalities, e.g., we are using (d - $d_{ACTM}$) as input the inversions instead of originally d and $d_{ACTM}$ separately. The $C_D$ and $C_{S0}$ and other infrastructures are also changed vastly. Part of the code is given below for 's'

```
Kount2 = 0
do l=1,lreg1              ! for number of regions, 84 in this case
  write(chl2,'(i2)') l
  do m=1,mtot1            ! for months
    do n=firstsrc,lastsrc    ! for years: firstsrc = 1998, lastsrc=2021
      kount2 = kount2 + 1
      ntime = nfirst + (n-firstsrc)*mtot1 +m-1
      src(kount2) = stemp3(l,ntime,1)
    enddo
  enddo
enddo
```

191: Similarly, what you have at the moment as matrices D_obs and D_ACTM should actually be vectors d_obs and d_ACTM, right?
183: change "prior source covariance matrix" to "prior source error covariance matrix"
184: change "data covariance matrix" to "data error covariance matrix"

All corrections are done

183-187: Some more detail needs to be given about how these Green's functions are created. Apparently, you are solving for monthly fluxes. Are you also averaging all the measurements together into blocks of one month, as well? Or are they treated at a finer temporal resolution? How far out in time are the Green's functions run? All 23 years, or across a shorter span? If truncated, how is the effect after that handled? Are the fluxes inside each emission region

divided by the flux uncertainty before being run through the transport model (so that the spatial distribution of the uncertainty inside the region is captured)? Or after the fact (i.e. uncertainty for the region as a whole)?

Replied above for the comment "169, Section 2.4:" The revised texts clarified these issues.

193: usually one uses the term "model data error" or "model data mismatch" to indicate that much of the error here is due to the model itself being unable to represent the data, as distinguished from a pure measurement error. That is not captured by your term "measurement data uncertainty".

Thank you for this suggestion. We have changed this to "model data uncertainty" here and all places in the manuscript.

Table 2 caption, line 2: change "Every PFU and MDU cases are" to "Each PFU and MDU combination case is"

Corrections are made

206-207: if you are multiplying by 3 and 4 in place of 2, shouldn't the ranges then become 0.3-3.0 and 0.4-4.0 PgC/yr? That is not what you give at the moment. Why do you change the lower bounds?

We stated the "maximum allowed" values. However, we agree with you that it is better to give the range, as given in the Table 2 already. Revised accordingly.

211: add a comma before "are used"
215: reword to "added these to an"

Corrected.

Figure 2: what does the subscript "pred" indicate? Are these the a posteriori results? Maybe something like "post" would be better...

233-234: Again, "posterior" or "a posteriori" would be more easy to understand in this context than "predicted", which could just as easily be thought to indicate the prior.

We had used "pred" for predicted flux. Following your suggestion all is changed to "post". This is now clarified in the Figure 2 caption. Thank you

In general, "FUR" is not a great statistic, since it depends heavily on the prior uncertainty, which can be made arbitrarily large and not change the final uncertainty much, at least in cases where most of the information is coming from the data rather than the prior.

Yes, we tend to agree with you, but we haven't been able to come up with anything different. So, we continue to use FUR.

201: Here you say that the PFU for the oceans in the control case is 1.0 PgC/yr, the same as it is in the fourth case, gpp_v4. However, in the left column of Figure 3, they appear to be different colors. Was the PFU for the oceans in the control case not 1.0 PgC/yr?

Apologies for this mistake. The PFU for the oceans in the control case is 0.75 PgC/yr. Text and Table 2 revised.

240: not the South Pacific -- a 1-5% reduction in uncertainty is not "good", I think.

We have removed South Pacific now. However, we think any measurable FUR change is a positive sign.

242: after "Northern", change "Africa. The Tropical" to ""Africa, and The Tropical"
244: add "on the" before "regional fluxes"? Otherwise, the meaning is not clear, so please clarofy
249: reword "into 1o x 1o spatial resolutions" to "to the 1o x 1o spatial resolution"

All of the above corrections are made in the revised manuscript.

253-254: You assert that the ensemble mean of the 16 different cases is the "best estimate", but how do you really know that this is the case? Maybe one of the looser prior cases is the best, because it allows the estimate to go closer to what the data indicate. Or maybe one of the tighter prior cases is the best because it damps down the dipoles caused by the generally underconstrained nature of these inversions. What criterion do you use to make this assertion?

We have now stated our criterion as (which is later shown in Fig. 5) :
"The best estimate criterion is based on closest agreement of the global total (FFC emissions + land and ocean sinks) fluxes with the global mean growth rate (section 3.2)."

There is no other observable quantity to validate inversion fluxes in a strict sense, and also used in GCP $CO_2$ budgeting process.

256-257: You should indicate what portion of the total uncertainty this ensemble-based measure pertains to. In particular, since you use a matrix inversion-based inverse method, you can presumably get a full-rank covariance matrix pertaining to the flux estimate (for each ensemble member). The uncertainties derived from this covariance would give you that portion of the total flux uncertainty due to the uncertainty in the measurements (the random error part) plus the uncertainty in the prior fluxes. The spread across the ensemble quantifies other errors -- say here what you think those are.

Yes, we have the full covariance matrix, but the regional fluxes we are analysing here do not conform with the inversion model regions. However, we have checked the a posteriori flux uncertainty for the TransCom sized regions are well over 2 PgC/yr. It is also clear from flux uncertainty reduction (FUR) statistics that the uncertainty for 84 inverse model

regions is not very large. Since we start with large a priori uncertainties (say, compared to TransCom Level 2 inversions), our a posteriori uncertainties are large.

That's one of the reasons we have performed an ensemble of inversion to assess the physically meaningful (can be questioned) uncertainties for regional fluxes. We have added these sentences in the article for clarification. "The regional and global land/ocean flux uncertainties estimated from the 16 ensemble members cover those arise from priori flux distributions, PFU, MDU. The uncertainties due to data coverage and model transport errors are not assessed here."

260: reword "3-dimensional CO2 observations" to "3-dimensional CO2 mixing ratio fields"?
Because you don't have an observation at each point in the full 3-d field.

Corrected.

262: You need to give a reference to the source of this data. In the References, you have a Schuldt et al reference pointing to obspack_co2_1_GLOBALVIEWplus_v7.0_2021-08-18. Does that pertain to this? Which did you use, v6.1 or v7.0? Please clarify.

Sorry, for missing the citation. Schuldt et al., 2021 is added for v6.1. The reference list is corrected accordingly.

271: "latitude intervals"?

279: Please indicate the total number of routine NOAA aircraft profile sites or time series you use. Table S4 seems to indicate that more than just these three sites were used. Maybe point to this Table S4 here in the text.
308: subscript "CO2"

Corrections and additions are made. We use 16 routine NOAA aircraft profile sites.

309: What errors do you mean to include in the term "uncertainties in the predicted flux"? Just those due to random errors (since uncertainty usually pertains to those errors)? If you mean to say "errors" instead of "uncertainties", then wouldn't some of those errors already be due to transport errors?

Yes, some errors would come from transport error, but as we have mentioned in the previous sentence the MIROC4-ACTM transport is validated for inter-hemispheric transport and transport of species in the upper troposphere and lower stratosphere using multiple tracers. Thus, we believe the biases and RMSEs will decipher mostly about flux errors.

We have revised this sentence as "Model transport is one of the sources leading to uncertainties in the predicted fluxes, but the simulations of $SF_6$ and age of air confirm the low transport error in MIROC4-ACTM (Bisht et al., 2021; Patra et al., 2018). Hence, the magnitude of biases and RMSE indicates predominantly the accuracy of the predicted fluxes (the errors due to model transport and measurement network are not explored in this study)."

321: "though" -- is this the word you want? The sentence, as it is written now, is unclear. Are you trying to say that the posterior results make reasonable corrections regardless of which prior they start from? Please reword so that this is clearer.

The sentence is revised as "The a posteriori results make reasonable corrections regardless of which a priori fluxes they start from, e.g., the gc3t case with net-zero annual flux or the 'gvjf' case with strong sink."

333: "However, the degree of freedom of our inversions is similar to the gridded inversions when spatial flux correlations of greater than 1000 km are assumed (Peylin et al., 2013)."

A gridded inversion with a correlation length of ~1000 km would have, say, 36x15=480 independent regions being estimated, more or less, compared to 84 in your case. This is not really comparable. I would agree, maybe, if you said ~2000 km. But what gridded inversions are using ~2000 km resolution? Please reword this to make your meaning clearer.

Revised as "The degree of freedom of our inversions is a few times smaller than the gridded inversions when spatial flux correlations of 1000-2000 km are assumed".

340: "two combinations": It appears that all 16 combinations of priors/prior uncertainties are shown in Figure 5 -- who do you say only two?

Revised for better clarity as

"Figure 5 shows the trends and interannual variability in the global fossil fuel (FF) emissions (used as input for the inverse model), land-biosphere, ocean, and annual atmospheric $CO_2$ growth rate for 16 inversion ensemble members based on two combinations of land-biosphere and ocean prior fluxes (VISIT and CASA for land-atmosphere, and TT09 and JMA for sea-air) and eight combinations of prior flux/data uncertainties (PFU and MDU)"

349-350: If you say that the uncertainties for the global land and ocean fluxes are 1.4 and 0.7 ppm, respectively, it makes me wonder whether you have accounted for the correlations (the off-diagonal terms) in the a posteriori covariance matrix properly in computing the uncertainties for those two regions. Other global inversions of in situ CO2 data have found the uncertainty for the global land flux to be down around 0.5 PgC/yr. Do you consider the off-diagonal terms in the a posteriori covariance matrix when calculating these uncertainty values on the global land and ocean regions?

Yes, the off-diagonal terms are included. Note that our a priori flux uncertainties are much greater than those used in TransCom studies for example. We use flat 2 PgC/yr for land and 0.75 PgC/yr for oceans in the control case.

Figure 5 caption, line 150: "brackets"

Corrected.

Figure 5 caption, line 150: "Numbers in the bracket in the legend are budget imbalance between inversions and observed CO2 growth rate." The description given here and in the text (lines 360-361) does not make it clear how these values were calculated. Do they measure the difference in _trend_ across the twenty years? (I.e., the difference in the beginning and ending values, divided by the number of years.) Or is it not the trend but rather the absolute offset that you are calculating? Or is it the RMS difference between individual annual values? Or monthly values? What are the units? Please do a better job describing this quantity in both places.

Mean of absolute offsets are given in PgC yr$^{-1}$. We have clarified it at both places as per your suggestion

373: "-induced changes": this doesn't work with a long parenthetical expression squeeze in between the original word ("La Nina") and this phrase. Please put the information inside the parentheses elsewhere (maybe in the caption to Fig. 5).

We have revised as per your suggestion. Parenthetical expression moved to Figure caption. Thank you.

377: "generally showing an increased ocean sink during strong El Ninĺ o events (e.g., during 2015-2016)". But your Figure 5c does not show this: it has an increased ocean sink at the end of 2016/beginning of 2017 and a reduced ocean sink in 2015. The 2015/2016 El Nino began in mid-2015 (or earlier) and was well over by mid-2016. The increased uptake, due to the capping of the thermocline in the East Pacific that occurs during the El Nino, should therefore be seen a full year before it is seen in Figure 5c. Please remove this or do a better job explaining what you mean.

We have deleted this part of the sentence. Such inconsistency arises from the lack of sufficient measurements in the Tropical Eastern Pacific region.

382: reword "caused by increasing pCO2 between the" to "caused by the increasing CO2 difference between the"

Done.

384: "and the gradual sink increase...": Wait, if you remove the strong increase in sink lasting up to 2012, possibly caused by the incorrect reporting of Chinese FFC use, then there is no increase in sink after that, but rather a decrease in sink (after 2012). Which effect do you want to argue for most -- the FFC effect or the CO2 fertilization effect? (It does not seem that you can have it both ways...)

Practically both are happening here. The FFC error is affecting flux estimation for a short period of 2001-2009, while the CO$_2$ fertilisation is slow but lasting process. We have made the specific period of FFC effect clear in the manuscript.

Figure 5d: With your sign convention for land and ocean fluxes, the quantity plotted here should be labeled "FF + (land+ocean)" -- i.e. change the minus sign to a plus sign.

Done.

398-402: This is really worded poorly and makes it difficult to understand what point is trying to be made. Really you are first giving the values the VISIT prior has for certain regions, followed by what the final predicted values are. However, it reads as if you are first giving the difference between the VISIT and predicted values (actually, it is not clear at all what the values in parentheses refer to). Please reword it to say: here is what the VISIT prior says the values should be, then here is what the predicted value is, then say where the final uptake is more or less than the prior. I.e., reword it for clarity.

Thank you very much for suggestion. We have revised the sentences as "Significant differences are seen in between a priori VISIT fluxes and a posteriori fluxes over Russia, East Asia and Europe. The VIST prior suggest the mean values of land uptake -0.76, -0.55 and -0.54 PgC yr$^{-1}$, respectively for Russia, East Asia and Europe; however the ensemble inversion suggest the ranges of fluxes from -0.33 to -0.37, -0.42 to -0.57 and 0.08 to -0.09 PgC yr$^{-1}$, respectively. In general, the inversions suggest substantial uptakes …"

406: "neighborhood"
408: "less certainly"
409: "groups"

Corrected.

411: since a sink of -0.18 PgC/yr could also be considered "mild", maybe change the wording here from "show a mild carbon sink" to "show almost no carbon sink"

Done.

412: Why do you mention that the VISIT prior has strong sinks over all three South American regions? Are you contrasting it to something? Not clear why you mention it.

Revised as "VISIT prior consists of strong sinks over all three South America regions, and for all the regions the inversions moderated the sinks and thus producing fluxes closer to the inversions using CASA prior even though the regions have no measurement sites"

418-419: It is not clear why you tie the trend towards increasing sink in East Asia to the trend in increasing FFC values there. If you are implying that the prior FFC numbers are overestimated there, please say that, to be clear.

Revised as "The predicted land carbon sink over East Asia tends to increase is tied to a rapid increase in FFC", and further explanations are given in the next sentences.

420-422: "Because the atmospheric data constrain the total net surface flux, the rapid increase in fossil fuel emissions is required to be compensated by increasing the natural land uptake of similar magnitude through inversion."  This compensation is only required if the atmospheric CO2 amount is not increasing to take up the fossil fuel added.  There is no requirement for local land uptake in areas of increasing fossil fuel input, since the winds can blow the input around across the globe quickly.  Please reword this to make your argument clearer.

Following your suggestions, we have revised this sentence as "Because the atmospheric data constrain the total net surface flux regionally when fluxes are constrained by observations, a biased high increase in fossil fuel emissions is required to be compensated by a biased high increase in the natural land uptake by inversion. If absolutely no constraints by observations, the compensation will occur in the regions where the biased FFC signals are transported by the prevailing winds."

428: "support"

430-431: reword "while the prior flux consisted no" to ", starting from a prior flux that has no"

435: change "due to" to "given by" or "caused by the assumed"?

437: add "in the" before "gvjf inversions"

Thank you very much for pointing out these corrections. All corrections are made.

437-442:  In order for this discussion to be understood better by the reader, you should mention that the incomplete measurement constraint in the inversions permits "dipoles" of flux errors to appear between neighboring regions (compensating errors of opposite sign due to the inability of the measurements to completely localize the source or sink in the right place), and that that is what is likely being seen here.

Thank you. We have borrowed your words and added a sentence here "These features appear likely because of the incomplete measurement constraint in the inversions permits "dipoles" of flux errors to appear between the neighbouring regions (compensating errors of opposite sign due to the inability of the measurements to completely localise the source or sink in the right place)."

443: replace "two-fold" with  "a two-fold higher"

444: replace "Inversion largely follows" with "The inversion results largely follow"

446: replace "as" with "is"

447: replace "of" with "off"

448: "is also known to have" -- what, "occurred"?  Please reword so that this makes some sense.

448-449: replace "tighter constrain by" with "a tighter constraint due to the"

450: replace "; while, we have" with ", even though we have"

Figure 8 caption:  it is unclear what "TDI calculation" refers to -- please spell out "TDI" and describe better what is meant by it here.

All corrections are made, and "TDI calculation" is replaced by "inversions" in Figure 8 caption.

462-465: This sentence needs to be reworded for clarity. It is only dimly clear what point is trying to be made, at the moment.

Revised and one sentence is added for clarity,

"The correlations were less than 0.3 between "gc3t" inversion and "gvjf" prior, which can be inferred as only some of the interannual variabilities were present in the gvjf prior, and the interannual flux variability for gvjf inversions are significantly different from gvjf prior. These results imply that the VISIT land ecosystem fluxes and GFEDv4s fire emissions inadequately represent $CO_2$ flux signals that are observed at the 50 measurement sites in our inversion."

474 and Table S3 caption: subscript "CO2"
Table S3: You need to give some more detail here on what ENSO index you are using when doing this correlation.

Corrected and ENSO index information given

470-471: "The CO2 flux anomalies in the tropical regions are strongly correlated with the ENSO index, while temperate and boreal regions are weakly correlated". This is an overly-generous characterization of the correlations you show in Table S3: there are only a couple regions that might at all be considered to have "strong" correlations with the ENSO index (Southeast Asia at +0.61, Western Pacific at -0.62), and this is only because that correlated variability was present in the prior at a slightly stronger level. Notably, the other set of priors did not give posterior estimates for these regions with a correlation stronger than 0.3. You are blithely twisting your narrative well beyond what the data justify.

Correlations are about 0.3 or greater for Brazil, Temp S America, Northern and Central Africa and Southeast Asia, as given in Table S3, for the gc3t inversion case which had no interannual variability in the prior flux, both for land and ocean. Also for these regions and gvjf inversion case, the correlation between MEI and posterior fluxes remained similar or slightly increased compared to MEI and prior fluxes.

We have now provided P-values as a significance test of the correlation coefficients in Table S3.

476: Russia is not one of the regions given in Table S3 -- maybe change to "North Asia"?

This was an overlook. Yes, North Asia – now changed to Russia

483: Figure 7 refers to ocean fluxes. Do you mean to point to Figure 6 or 8?

Yes, it should be Fig. 8 (or Fig. 6). Fig. 8 is now cited.

492: In your discussion of the large IAV seen in Oceania, you do not mention that this is all coming from the gvjf prior and not from the data. This is because the a priori flux uncertainty for that region is quite tight, according to Figure 3a (except for the control case -- why is the uncertainty in the control case so much higher there than for the other prior

cases? Is this an error in Figure 3a?). Because the fluxes for the two different prior models (gc3t and gvjf) are so different, it would have been more reasonable to have used a looser prior for this region, reflecting the disagreement between the two actual prior timeseries that you used. I like your discussion of the variability in the GFED prior, but it is unfortunate that you did not leave the fluxes for this region loose enough to test whether this prior is in fact in agreement with the available CO2 data.

We actually have the inversion cases of ctl_ux2_gvjf & ctl_ux4_gvjf, which are clearly suggesting some differences from the prior by the inversions (Fig. 6o). But some part of the Australian landmass is weakly constrained by observations (Fig. 3). In general, our inversion suggests some consistency in the $CO_2$ flux IAV for gc3t and gvjf inversions (r=0.43), but the flux variabilities are much weaker for gc3t compared to those for gvjf prior or predicted fluxes.

why is the uncertainty in the control case so much higher there than for the other prior cases? :
In the control case we used fixed 2 PgC/yr PFU for all land regions, but in the gpp_v* cases the PFU are proportional to GPP of the region, which is low for Australia due to the lack of dense biosphere.

We believe more targeted research is needed to answer all the important questions you have raised. Thus, we are not changing the discussions here, for not to be too speculative.

500: You seem to be contrasting the gc3t and gvjf priors here -- please add something like "The gc3t" at the beginning of the sentence to indicate that you are talking about that case first, before switching to talk about the gvjf case.

Thank you. Done

502-504: "The oceanographic observations indicate that sea surface temperature and pCO2 in the equatorial warm pool areas (5°N–5°S, west of the dateline) are not sensitive to El Ninİ o conditions (Takahashi et al., 2003)." If that is the case, how do you explain the "strong" correlation in the West Pacific in the gvjf case, both in the prior and final estimate? What about the JMA model is correlated with ENSO if not SST and pCO2?

We have added this discussion here:
"The oceanographic observations indicate that sea surface temperature and $pCO_2$ in the equatorial warm pool areas (5°N–5°S, west of the dateline) are not sensitive to El Niño conditions (Takahashi et al., 2003), but a strong correlation is found for the West Pacific region in the case of JMA ocean prior that is driven by $pCO_2$ measurements and sea-surface temperature. The gc3t inversions did not produce expected (negative) correlation for $CO_2$ fluxes and ENSO index for the both East and West Pacific regions, due to the lack of observational coverage. Patra et al. (2005a) showed that the global ocean flux variability is significantly underestimated or even produced opposite phase for strong El Nino of 1997/1998, if the Pacific Ocean Cruise data are not used in inversions."

521-522: reword this first sentence so it is clear that the CASA model is the one with the July peak.

We have revised the sentence now as "Seasonal cycle amplitude for CASA prior flux for land total is 33.6 PgC yr$^{-1}$, and that for VISIT is weaker at 23.8 PgC yr$^{-1}$, and the peak of the growing season (when the net flux is most negative) occurred in July for CASA that is one month after the VISIT (Fig. 9, top-left panel)".

524: reword this to make it clear that it is the a posteriori, or predicted, estimates for the gc3t case that you are comparing to the prior.

527: It appears that you are still discussing the total land flux at this point, which is not shown in Fig 9a, but rather the figure to the left of that one -- please fix this reference here.

We have made several small corrections for clarity, based on these 2 comments.

534: change to "Northern land fluxes drive"

539: change "are" to "is"

Corrected.

539-542: You have described why the prior fluxes agree or disagree here, but not why the posterior fluxes do so. For the posterior fluxes, they do not converge well in the tropics mainly because of the general sparseness of data there, or rather data that constrain the fluxes there. Perhaps noting that, as well, would be useful.

We have added a sentence : "Posterior fluxes for the tropical regions also do not converge well mainly because of the general sparseness of $CO_2$ data (Patra et al., 2013)"

547: add "adjoining" before "neighborhoods'" to indicate that it is observations in the surrounding area that are providing the constraint.

552: add "and" before "East Asia"

560: add a comma before "caused"

All corrections are made.

563: "It is not easy for us to explain the mechanism for the Northern Ocean to be a weaker sink in summer than in winter." One possibility is simply the reduced solubility of CO2 in warmer waters leading to an outgassing of CO2 then.

Thank you for the suggestion, we have further scrutinized the Yasunaka et al. paper and added
"It is not easy to put forward a hypothesis for the weaker sink in summer than in winter of Northern Ocean, while we can speculate that the atmospheric $CO_2$ decrease in polar air exceeds compared to the decrease that occur over the surface sea-water and reduced solubility of $CO_2$ in warmer water. Indeed, Yasunaka et al. (2018) have shown that the Greenland-Norwegian seas and Barents Sea are indeed acts as milder sink of $CO_2$ (flux = -4 to -5 mmol m$^{-2}$ day$^{-1}$) during June-August compared to the October-March (flux = -10 to -15 mmol m$^{-2}$ day$^{-1}$), and the Chukchi Sea and Arctic Ocean

show strongest uptake in October. Thus, as whole the Northern Ocean of our study could act as the weakest sink in summer months."

568: add a comma after "Overall"

Figure 10 caption, 2nd line: replace "Each inversion cases" with "The different inversion cases"

Table S4 caption: change "is" to "are"; Also you need to say how you calculate the differences that are being plotted: is it model-observation? Is it the average of the a posteriori fluxes for all 16 cases that make up the modeled value?

Corrections and clarifications are made.

590-593: It is not clear what distinction you are making between the 25 and 75 percent error bounds. Aren't these just the two sides of the mean (i.e. 25% on either side of the mean, given by the bounds of the boxes in Figure 10)? When talking about the 25% results, do you really mean the 5%/95% bounds (given by the whiskers)? Not clear as currently written...

We revised this text as "Flux estimates for all the land regions remain quite uncertain, as seen from the 5 to 95 percentiles range of the 16-inversion ensemble (whiskers) at about 0.3 PgC yr$^{-1}$ for the land regions and typically less than 0.2 PgC yr$^{-1}$ for the ocean regions. The fluxes at 25 to 75 percentiles range show slightly reduced uncertainties – a large reduction is not seen compared to the 5 to 95 percentiles range because the two a priori models often formed two different sets of $CO_2$ flux values"

595: This lack of reduction for the larger regions makes me wonder again whether you have properly accounted for the off-diagonal terms in the a posteriori covariance matrix when grouping regions.

We have followed the TransCom formulation for this calculation. Usually, we have about 5 regions in one aggregated region. Here are the posteriori flux uncertainties for the TransCom regions (except that the Temperate Asia is broken in to South and East Asia):

| Region name | Flux_Correction | Flux_Uncertainty |
|---|---|---|
| Boreal N. America | -0.16 | 2.13 |
| Temperate North America | -0.96 | 2.79 |
| Tropical America | 0.43 | 3.20 |
| South America | -0.01 | 3.18 |
| Northern Africa | -0.09 | 3.22 |
| Southern Africa | -0.07 | 2.78 |
| Boreal Eurasia | -0.22 | 3.09 |
| West Asia | -0.43 | 3.81 |
| East Asia | -0.26 | 2.63 |
| Tropical Asia | -0.13 | 3.32 |
| Australia | -0.30 | 2.38 |
| Europe | -0.04 | 3.00 |

| | | |
|---|---|---|
| North Pacific | -0.11 | 1.29 |
| West Pacific | 0.00 | 1.00 |
| East Pacific | 0.20 | 0.91 |
| South Pacific | -0.09 | 1.08 |
| Northern Oean | 0.13 | 0.85 |
| North Atlantic | -0.12 | 0.92 |
| Tropical Atlantic | 0.03 | 0.93 |
| South Atlantic | -0.02 | 0.97 |
| Southern Ocean | -0.06 | 0.83 |
| Tropical Indian Ocean | -0.12 | 1.41 |
| South Indian Ocean | 0.03 | 0.84 |
| total | -2.39 | 8.38 |
| total-land | -2.26 | 8.20 |
| total-ocean | -0.13 | 3.38 |

We have now revised the sentence as "each of the 15 land analysis regions have predicted flux uncertainties in range of 2.1 (Boreal North America) to 3.8 (West Asia) PgC yr$^{-1}$ for the control gc3t case, as the reduction from prior flux uncertainties were small by inversion for most region (Fig. 3)"

Sorry for not being precise in the submitted manuscript.

615: "hosts" and "and hence is"

624: it is not clear what you mean by "at a higher magnitude" -- please reword for clarity.

626: put the wiggle on the n in "El Nino"

633: "unanimously" doesn't seem to be used correctly here -- remove it?

636: subscript "CO2"

640: "is in the North Pacific,"

641: instead of "CO2 uptake rate", say "change in CO2 uptake", since it is not very clear that by "uptake rate" you mean the time derivative of uptake.

644: the Long et al reference is missing from the Reference list -- add it

Thank you very much for these suggestions. All the corrections are made.

646. This new section should presumably be numbered "6.", not "4.", since it follows "5.", and the Conclusion section later as "7.", not "5."

All the sub-sections in the Results and Discussion section are numbered as 3.x for simplicity, and the Conclusions as '4'.

649: You need to define how you came up with these three sets of fluxes: 'gc3t', 'gvjf', and 'ensm' – are they created from the average of the 8 gc3t and 8 gvjf ones, and the average of all 16? If so, say so.

We have revised the text as "three sets of prescribed fluxes: "gc3t" (case: ctl_ux4_gc3t in Table 2), "gvjf" (case: ctl_ux4_gvjf), and "ensm" (average of all 16 inversions)."

651, 653: "ATom"
Fig 11 caption, line 1: "meridional"

Thank you. Corrections are incorporated in the revised manuscript.

Fig 11 caption: you should indicate which quantity is subtracted from which when computing the biases -- it is not clear from the figure.

"model-observation bias" is now mentioned.

664: "Most of the aircraft data over these latitude bands are available over the continental regions, and this comparison suggests a higher sink than the estimated sink by inversion."

It is not clear whether the aircraft data that you refer to here are the ATom and HIPPO data that you were discussing in the previous sentence, or other data. Since the sign of the observation-model difference has changed, this implies that you are discussion some other set of data. Please clarify this. If the data is still the HIPPO and ATom data, then the two sentences seem to contradict each other. Please reword these sentences so that your meaning is clear. Also, in the final sentence in this paragraph, why do you say that the models seem to do a good job in terms of the mean CO2 level when in the previous two sentences you have just pointed out that they do not do a good job (i.e. they are biased), at least in the north?

Sorry for the unclear discussions. The text is revised now as
"The NOAA aircraft observations show a high bias during boreal summer throughout the troposphere over the US and Canada, implying possible seasonally dependent errors in posterior fluxes over these latitude regions (Fig. S7). When the aircraft data is over the high latitude continental regions, model-observation comparison suggests a stronger surface $CO_2$ sink is estimated by inversion compared to what is suggested by vertical profile gradients. HIPPO for the month of July also show negative model-observation mismatches near the surface (Fig. S6). But the mismatches turn positive in the higher altitudes, above about 1 km, and thus the model and observations averaged over 0-2 km are in much closer agreements (Fig. 11c). Based on these comparisons, the simulations from the ensemble mean of 16 inversion cases ("ensm") show lowest mean bias, in comparison with gc3t or gvjf inversions, and suggested to be most suitable flux estimation for quantifying the global land and ocean carbon sink on the timescale of annual mean and its decadal trend."

673: "The inversions underestimate"

Done

693: It is not clear what the broken lines are meant to indicate in Fig 12d-f. Are these what you get using the prior fluxes, and the solid lines what you get using the predicted fluxes? Please reword this both in the text and in the caption to Fig 12, so that this is clear.

Figure caption and text revised according to your suggestions.

694-697: "In the case of predicted data, the inversion fits the observation well due to minimisation of prior model-observation differences, but when the simulations are run using predicted fluxes, the (small) systematic biases produce a (large) cumulative effect over the model integration period."

This is NOT a general feature of flux inversion models, but rather a peculiarity of your inversion setup. In most inversion models, when you do a forward run with the optimized fluxes, you get the same modeled measurements as the inversion would give (unless for some reason you choose to run the model at a different resolution than what was used in the inversion). What is it about your inversion setup that causes this not to be the case? One possibility that comes to mind is that you have not extended your Green's functions runs out in time long enough: how long do you run them for? How do you handle the influence of a Green's function after this (i.e. after the end of your run)? You must provide more discussion on why you get different modeled measurements from what you assume in the inversion when you run the optimized fluxes forward through the model.

It is now given clearly in the Inverse method (section 2.4) that the Green's functions are run for 4 years. We have checked that the pulse signals are homogenously distribution at the end of 48 months, and we believe further extension of the simulations are not needed. But it is something we should test in the future by running the Green's functions well beyond 4 years.

However, following suggestions from you and reviewer#1, we have deleted lines 692-720 from the submitted version of the manuscript. Also deleted are Supplementary Figure S10, and the final paragraph from the Conclusions. We hope these actions will get rid of much of the confusions, as mentioned here and in the comments below.

"..when you do a forward run with the optimized fluxes, you get the same modeled …"

"You must provide more discussion on why you get different modeled measurements"

Fig 12 caption and legend: it is not clear what the dashed lines labeled 'gc3t' and 'gvjf' indicate -- are these the modeled measurements given by these two priors? Please say in the caption what they are. If they are the modeled measurements given by the priors, why do you not also plot these lines for the top panels?

699: "We speculate that MIROC4-ACTM produces stronger sinks in the high northern latitudes":

stronger than what? Please reword this to make the meaning clear.

697-707: "It is also interesting to note that the meridional gradients in biases for independent aircraft observations (Fig. 12a,b,c) and sites used in inversion (Fig. 12d,e,f) show opposite phases, i.e., most negative and most positive at 25oN, respectively. We speculate that MIROC4-ACTM produces stronger sinks in the high northern latitudes (negative model-observation bias at surface sites over 75oN or HIPPO/ATOM latitude-altitude plots in Fig. S5, S6), which can arise from the model's inability to simulate the sites over the land because of the coarse horizontal resolution. Thus, resulting in a weaker sink or a stronger source in the northern tropics and subtropical (25oN) regions, respectively. The tropical source is then transported to the mid-high latitudes, which is captured by the aircraft observations, as a positively biased concentration. This experience suggests a need for new forward model simulations using inversion fluxes, not the optimised atmospheric $CO_2$ fields during data assimilation, should be used for evaluating inversion fluxes with the help of independent observations."

This discussion is not clear and makes no sense to me. Why should 75 deg N be an important inflection point for the surface data (there being very few surface sites that far north, anyway)? If there is a stronger sink than there should be in the northern extratropics, then yes, there could be a balancing stronger source south of that. But how could the positive perturbation in atmospheric $CO_2$ then jump over the negative perturbation to the north of it to then somehow cause the positive model-obs differences seen in the far north (Figure 12 and S5)? And even if this were a plausible explanation, how does this relate to running the optimized fluxes back through the forward model? An alternate explanation would be too-weak mixing during the summer and too-strong mixing during the winter in the north, causing overestimation of the summer drawdown and underestimation of the winter accumulation of $CO_2$ in the PBL.

710 and Figure S10: If the same transport model is being used for the forward run as was used in the inversion, and run at the same resolution, then why would you expect that it would give a different simulation of the 3-D $CO_2$ field than was obtained in the inversion? What is the underlying reason? (I can think of one possibility: that the Green's functions used in the inversion were not run out far enough in time, driving basis function time truncation errors in the inversion. Is this the reason?) Please do a better job describing why you think doing a final forward run would give different modeled $CO_2$ fields, if this is a perfect model situation and the same model is being used for the forward run as in the inversions.

711-720: This whole discussion also makes no sense to me. For $CO_2$, a model with weaker interhemispheric transport causes a stronger N/S gradient when forced with NH-dominant fossil fuel emissions. When compared to the weaker observed N/S $CO_2$ gradient, this then requires a stronger NH $CO_2$ sink than a model that gives a weaker N/S $CO_2$ gradient. It is not very complicated and "complex interactions" need not be invoked. I agree that one should not use the assimilated data as a test, but rather comparison against independent data. But you do compare against independent data here (HIPPO, ATom), so why do you need this whole paragraph in the first place. Please do a better job with your argument, so that the reader can understand your point.

We believe the final two paragraphs are not clear and appearing to confuse even the expert readers. With that in mind we have decided to delete these two paragraphs, Supplementary Fig. 10, and the final paragraph in this revised manuscript.

Regarding the final paragraph before Conclusions (lines 709-720), it is nice that we have a general agreement on how the inversion estimated fluxes are to be tested, i.e., by comparison against independent data. As the reviewer has kindly pointed out we have already done both comparisons with independent flux results from RECCAP and aircraft observations to assess our inversion results, and this paragraph and Figure S10 are redundant.

723: You should be more specific and say that the land and ocean absorb 53% of the FFC fluxes, not of the total anthropogenic fluxes, because if you add in deforestation (which is an anthropogenic flux), it is no longer 53%.

This sentence is revised as "The terrestrial biosphere (2.58 PgC yr$^{-1}$) and ocean (1.54 PgC yr$^{-1}$) absorb about 46% of the emissions due to fossil fuel and cement production (8.9 PgC yr$^{-1}$) in the period 2001-2020."

730: add a comma before "and two"
734: replace "resultant" with "result"

Corrected.

735-736: "The spread between the ensemble members provides us a reasonable measure of the inversion estimated flux uncertainty but lacks the quantification of transport model uncertainty."

It seems to me that the spread in the ensemble results should quantify the variability due to only those things that are varied across the ensemble: prior fluxes, prior flux uncertainty, and characterization of the MDU. It should not be expected to capture the usual estimation uncertainty due to errors in the measurements and errors in the prior flux (why? because the spread across the ensemble only quantifies the effect of mis-characterizing or changing the assumed statistics for those quantities, but does not capture the uncertainty due to those errors themselves). Therefore, in addition to the errors due to transport, you should also add on these usual estimation uncertainties to get the total errors. This would be a good place to mention that additional error source.

This sentence is revised as

"The spread between the ensemble members provides us a reasonable measure of the inversion estimated flux uncertainty but lacks the quantification of the roles of transport model uncertainty or the inherent errors in the measurements and the prior fluxes."

742: replace "extratropical" with "extratropical southern", since you are focusing only on the south not the north

Done

It is unclear what feature in the flux results you are referring to here, with this statement. Please say what flux feature you are discussing -- global total? global land total? global ocean total?

We have revised this as "The ensemble of inversions splits into a "near-neutral" group and a "strong-source/sink" group based on the priors for the tropical and extratropical southern land regions."

750 remove the comma before "in less agreement"
752: "ATom"
766: what do you mean by "unanimously"? That it is true across all 16 cases?
772: "North Pacific"

Thank you for these suggesting these corrections. All of the above corrections are made in the revised manuscript.

772: What do you mean by "the most considerable CO2 uptake"? The uptake in the Southern Ocean that you discuss here is not as large as the uptake in the land regions you just mentioned. Do you mean "the most considerable CO2 uptake in the oceans"?

Revised as "North Pacific with a mean flux of -0.55±0.05 PgC yr$^{-1}$, and also considerable $CO_2$ uptake is estimated for Southern Ocean, where $CO_2$ uptake increased from -0.12±0.07 PgC yr$^{-1}$ in 2001-2009 to -0.33±0.06 PgC yr$^{-1}$ in 2010-2019"

778-779: "There is no doubt that this set of results is unique because they close the year-to-year budget of decadal CO2 changes in the atmosphere."

Almost all inversions close the year-to-year budget in decadal CO2 change, due to the strong observability of the fossil fuel input minus atmospheric increase. Given that, why is your set of results unique? I have the little doubt that it is not. Please reword to make your point clearer.

779-780: "The bottom-up inventory or other modelling system still has limitations in closing year-to-year budgets."

You have used two sets of priors here that make no attempt to satisfy the long-term CO2 trend in the atmosphere by trying to model an appropriate global land biospheric uptake. That does not point to a limitation in the modelling systems but rather a deliberate choice that you have made in the work you present here.

We have deleted the final paragraph of Conclusions in the revised manuscript, following these comments from you and Reviewer#1

---

## Author Response (AR2)

**Reply to Review #2**

I thank the authors for their work addressing my comments. I believe that the text is clearer now and that the reader will have a better opportunity to understand the work done here and its significance. While the authors have generally done a good job addressing my questions, I still have some worries and questions about the work that I would like addressed before giving the thumbs up for publication.

Thank you for the helping us to improve the manuscript.

In my original comments (at line 595) I questioned the small uncertainty reductions (under 10%) seen in Figure 3 for many regions, especially those in the tropics, and speculated that the uncertainties for the grouped regions might not be being calculated correctly.

In Figure 3, we are showing uncertainties for individual 84 regions (used in the inversion), not for the grouped regions. We plotted the monthly flux uncertainties, averaged over the analysis period 2001-2020, for both a priori (as input to the inversion model) and a posteriori (o/p of the inversion) fluxes. It is not uncommon to find low uncertainty reduction by inversion based on the Bayesian a posteriori flux uncertainty for each inversion regions (e.g., Gurney et al., 2002), and more importantly there is "difficulty" to calculate Bayesian posterior uncertainty for most inversions accurately/appropriately (we have faced this since working on Thompson et al., Nature Comm, 2016, and also in GCP Global carbon budget analysis, series of ESSD papers). Due to such difficulties, we have developed the ensemble inversion approach by accounting uncertainties arising from prior fluxes (CASA vs VISIT land, Takahashi vs JMA ocean) and input parameters of the inverse model (PFU, MDU).

In light of your comments, we have added a few lines of clarifications in the main manuscript:

Section 2.5: Performance of inversion using a posteriori uncertainty
"The inverse model output monthly means flux corrections and a posteriori flux uncertainties for each of the 84 regions, and the full error covariance matrix of dimension 24192 × 24192 (=84 regions × 12 month × number of year). The monthly time and spatial covariances are accounted for flux uncertainty calculation when annual mean values are calculated for aggregated regions or global budgets. In the aggregation scheme, the larger regions have to follow the boundaries of 84 regions, contrary to the method proposed in section 2.6 by using ensemble inversions where ensemble spreads can be calculated for any region of interest.

We use flux uncertainty reduction (FUR, in %), based on the mean values without time aggregation …"

Section 3.2: in the final sentence
"The ensemble spread is much lower (Table 3; MIROC4-ACTM columns) compared to the inversion predicted flux uncertainties, which are in the range of 1.4 and 0.7 PgC yr$^{-1}$ for the global land and ocean, respectively, even after accounting for the monthly time and spatial covariances (vary from low values of 0.8 and…"

Similarly, in my comment on lines 349-350, I suggested that the 1.4 PgC/yr uncertainty on the global land flux was a few times higher than it should be. The authors, in their response, gave a list of a posteriori uncertainties at the scale of Transcom regions (lines 113-142 of the response) that, rather than

squelching my worries have instead exacerbated them. Along with the uncertainties for the individual Transcom regions, they give uncertainties for the global land and global ocean total, along with the global land+ocean total: those for land and the land+ocean total are up around 8 PgC/year -- those values seem to be in conflict with the 1.4 PgC/year value given in the paper, so which is correct? If these uncertainties are indeed up around 8 PgC/year, that would suggest that the correlations in the a posteriori covariance matrix are not being considered (since taking the sum of the squares of the uncertainties for the individual Transcom regions given in lines 113-142 of the response gives about (8 PgC/year)^2).

Here we calculated the aggregated fluxes and flux uncertainties for annual mean values (the covariances accounted for) before taking the long-term means which are given in the text. Thus the 1.4 PgC/yr uncertainty is correct for the annual mean fluxes (temporal covariances accounted).

We have rechecked the calculation of aggregated region flux uncertainties. In the reply, what we produced was for monthly flux values – only spatial covariances accounted for but not the temporal covariance. We believed those a posteriori values, without accounting for temporal covariances, are more relevant to compare with the a priori flux uncertainties (as is done for FUR calculation).

The 1.4 PgC yr$^{-1}$ uncertainty on the global land flux is several times higher because we use much larger prior flux uncertainty compared to inversion like TransCom. This was already discussed in the manuscript as follows (line#406-409)
"The ensemble spread is much lower (Table 3; MIROC4-ACTM columns) compared to the inversion predicted flux uncertainties, which are in the range of 1.4 and 0.7 PgC yr$^{-1}$ for global land and ocean, respectively (vary from low values of 0.8 and 0.5 PgC yr$^{-1}$ for gpp_v2 cases to 1.6 and 0.9 PgC yr$^{-1}$ for the gpp_v4 inversions)."

For "gpp_v2", the uncertainty is as low as 0.8, close to the commonly reported values. Note that in gpp_v2 inversion PFUs for some regions are up to 2 PgC/yr, which is still higher than the TrransCom inversions.

We hope this clarifies your doubt. Sorry for the confusion.

Another possibility is that there is a problem with the posterior covariance matrix that they are using for the calculation: for example, if the correlations given by the off-diagonal elements were computed incorrectly. In my original comments (lines 183-187), I asked about one point that might lead to that covariance matrix to be calculated incorrectly: if the Green's function relating fluxes at a given time to mixing ratios at later times were to be truncated too soon. The authors responded that they run the Greens' functions for each flux pulse out for four years. This is long enough to capture the spread of the input pulse to the point where it have negligible latitudinal gradients and is typical what has been done in previous inversions. However, what the authors do not address (and what could cause problems that might result in an incorrect covariance matrix) is what is assumed for the influence of those fluxes at times after those four years: is zero influence put in the matrix (bad), is the fully spread-out value of about 0.4 ppm / (PgC/year) put in (better), or some exponential decay to the spread out value (even better)? The authors point to the original Rayner et al code that they have modified for use here, but don't explicitly address this issue. If they put zeroes in matrix J for all years after Year 4 instead of a better spread-out value, I could see how the correlations in the posterior covariance could be too low and the a posteriori uncertainties would be wrong. The original Rayner et al code may have been used for only a short span (four years) such that this point would not have been an issue. As things stand at

the moment, I do not have confidence that the posterior covariance, upon which so many of the uncertainties discussed here rest, is being calculated correctly.

Greens functions decay nicely to the common value after 47 months in MIROC4-ACTM forward simulation of the monthly pulse functions, and we have kept it constant afterwards in the inversion code. We have not applied "exponential decay to the spread-out value", but that will preassembly of minor importance for $CO_2$ as it has no chemical loss present in the atmosphere. We only anticipate minor decay after four years due to the mixing through the whole atmosphere by slow transport from troposphere to stratosphere.

As stated earlier, we think the flux uncertainties are calculated correctly but the presentation varies based on whether or not temporal covariances are included in the annual/long-term mean flux uncertainties.

In Section 2.4, we add following line for clarification (line #211).
The elements of **J** for later months are kept constant at the value of 48th month.

Another issue about which I asked for clarification in my original comments (lines 70-81 and 694-967) was why an additional forward run with the optimized fluxes was needed (i.e., why the impact of the fluxes on concentrations through the J matrix did not fully represent the effect). The authors give some vague generalities but do not explicitly say what the answer is. They also delete some text. However, it is possible that the answer (not given) is that there is no influence whatsoever after four years from a given set of fluxes (i.e. zeroes in the J matrix after Year 4). As mentioned above, this would have an impact on the a posteriori covariances calculated.

We are not using all the measurements in the inversion system (only 50 sites are used), as measurement data gaps produce artifacts in the flux interannual variability. The forward simulation of the fluxes is needed for validation of the a posteriori fluxes, in particular using the large numbers of aircraft (or satellite) measurements. (It is only in the data assimilation system that we can get full 4-D concentration field, but not in the case of batch inversion in our case).

We calculate and store the J-matrix only for selected fixed sites, which could be used in inversion. Thus when the a posteriori fluxes are to evaluated with independent measurements (those not used in inversion), we need to simulate the $CO_2$ concentrations using the inversion corrected fluxes. This is done for GCP-$CO_2$ submissions of MIROC4-ACTM (since 2018), or other multi-model assessments (Gaubert et al., BG, 2019; Long et al., Science, 2021).

Some of the texts were deleted for cleaning up the discussion in the first submission relating to further development of inversion evaluation metric, which was found to be confusing to both the reviewers. Hope that we are not missing any significant text here.

I would need to have answers to these questions before I will feel confident in the results presented here. Errors in the J matrix would affect not only the uncertainties calculated in this manuscript, but also the flux estimates themselves, and could materially change some conclusions presented.

As we have shown in several figures and tables that the inversions are able to come to common solutions of a posteriori fluxes from extreme a priori fluxes, we have a strong feeling regarding the validity of the

fluxes. Direct comparison of the ensemble means fluxes with IPCC-AR6 (Table 3) and RECCAP-1 fluxes (Table 4) also raise the confidence.